# Application of Micro/Nanomotors in Environmental Remediation: A Review

**DOI:** 10.3390/mi15121443

**Published:** 2024-11-29

**Authors:** Tao He, Shishuo Liu, Yonghui Yang, Xuebo Chen

**Affiliations:** School of Electronic and Information Engineering, University of Science and Technology Liaoning, Anshan 114051, China; ht18191863591@163.com (T.H.); 18872010432@163.com (S.L.); yangyh2636688@163.com (Y.Y.)

**Keywords:** self-propulsion, detection, degradation, environmental remediation

## Abstract

The advent of self-propelled micro/nanomotors represents a paradigm shift in the field of environmental remediation, offering a significant enhancement in the efficiency of conventional operations through the exploitation of the material phenomenon of active motion. Despite the considerable promise of micro/nanomotors for applications in environmental remediation, there has been a paucity of reviews that have focused on this area. This review identifies the current opportunities and challenges in utilizing micro/nanomotors to enhance contaminant degradation and removal, accelerate bacterial death, or enable dynamic environmental monitoring. It illustrates how mobile reactors or receptors can dramatically increase the speed and efficiency of environmental remediation processes. These studies exemplify the wide range of environmental applications of dynamic micro/nanomotors associated with their continuous motion, force, and function. Finally, the review discusses the challenges of transferring these exciting advances from the experimental scale to larger-scale field applications.

## 1. Introduction

Rapid industrialization has led to the release of significant quantities of harmful pollutants into both water and air resources. To protect human health and the environment, it is essential to implement effective pollutant treatment methods, utilizing biological, physical, or chemical processes. However, the effectiveness of these treatment approaches is often limited by diffusive mass transfer, which necessitates the introduction of additional agitation to improve their efficiency. The emergence of micro/nanomotors (MNMs) adds a new dimension to the field of environmental remediation [1]. Notably, self-driven MNMs that convert energy from their surroundings into movement demonstrate significant potential for addressing the diffusion limitations associated with treatment operations by using autonomously moving substances [2,3,4,5,6,7,8,9]. Their continuous motion facilitates effective mixing without requiring external agitation, thus minimizing cleaning time and greatly improving efficiency [10]. Since the initial studies on self-electrophoretic and H_2_O_2_-driven MNMs [11], researchers have explored various alternative propulsion mechanisms, including the Marangoni effect [12]; light-induced motion [13]; and electromagnetic [14,15,16,17,18,19,20], thermal [21,22,23,24,25,26,27,28], optical [29,30,31], ultrasonic [32,33,34,35,36,37,38,39], and biomixed motions [40,41]. The integration of efficient propulsion and motion control with easy functionalization, load pulling, swarming, and chemotaxis behaviors has enabled a diverse range of applications for these devices, such as drug delivery [42,43], analytical sensing [44,45,46], energy production [47], and assisted fertilization [48].

The unique features of synthetic MNMs have opened up new possibilities in the realm of environmental remediation. Recent proof-of-concept research has underscored the substantial influence of MNMs in this area, with applications ranging from water quality assessment to pollutant degradation and elimination. Wang’s research team pioneered the early investigation of MNM-based environmental applications, utilizing a motion-dependent Ag^+^ detection approach that capitalizes on the effect of pollutants on MNMs’ movement speed [49]. This pioneering research lays the foundation for future MNM-based sensing protocols. Additionally, the researchers showcased a case study utilizing functionalized tubular MNMs for the real-time removal of oil droplets from water [50]. Later, Sanchez and Schmidt applied rolled-up Fe/Pt tubular MNMs to break down organic pollutants in water through the Fenton oxidation process. These early efforts marked the first proof-of-concept demonstration of MNMs in water purification applications [51]. The number of studies exploring the environmental applications of MNMs has grown rapidly in recent years. Many of these investigations have heavily depended on the use of H_2_O_2_ as a fuel source. Although H_2_O_2_ is commonly employed as an oxidizing agent in water treatment and gradually decomposes naturally, its widespread use may face limitations. Thus, it would be prudent to focus on developing MNMs powered by water or exploring eco-friendly, fuel-free propulsion methods like light or ultrasound [52,53]. Recent efforts have simultaneously aimed to improve the effectiveness of MNM-based remediation protocols by introducing advanced nanomaterials, such as graphene. However, several key challenges must still be tackled before these breakthroughs can be successfully implemented in real-world environmental applications. For example, further investigation is needed to resolve the problem of large-scale contaminated site remediation and to conduct more comprehensive toxicity assessments of pollutants.

This review highlights the progress and potential of self-propelled MNMs in environmental applications (Figure 1). First, it demonstrates how the mobility of functionalized MNMs is revolutionizing environmental detection, offering key advantages such as reduced analysis times and real-time monitoring. Next, it explores the use of MNMs in detecting organic substances, degrading pollutants, eliminating harmful bacteria, removing oil droplets from water, and adsorbing contaminants. Finally, the main challenges hindering the development of MNMs into practical materials are discussed, along with possible future research directions in the field. These studies underscore the vast potential of mobile MNMs across various environmental applications, especially in dynamic remediation and sensing.

## 2. From Materials to MNMs

MNMs are micromachines designed to autonomously perform complex tasks or operate under hu man guidance. With dimensions under 100 µm, these microstructured materials can harness energy from their environment and convert it into motion [56,57]. Their versatility stems from the synergistic interaction between their dynamic capabilities and distinct physicochemical properties, which are influenced by their size, shape, and structure at the micro- and nanoscale. Moreover, MNMs exhibit superior performance in various applications, including environmental remediation, when compared to traditional static micro- and nanomaterials [9,58,59,60,61,62,63,64]. The active motion of MNMs triggers fluid movement around them, further enhancing the mass transfer efficiency of chemical reactions. This dynamic approach overcomes the limitations of traditional static methods, where chemical reactions rely solely on passive diffusion. As a result, MNMs can interact more effectively with waterborne contaminants, significantly reducing purification time. As the complexity of the intended function increases—such as capturing specific contaminants or utilizing a population of MNMs to boost removal and degradation efficiency [65,66]—the design of these micromachines becomes more intricate. By incorporating propulsion, multifunctionality, environmental adaptability, collective behavior, and intercommunication, conventional micro- and nanomaterials are transformed into intelligent MNMs (Figure 2). Achieving self-propulsion requires robust fabrication techniques that create structural asymmetries, disrupting the local field symmetry to enable motion [67,68,69,70,71,72] (Table 1).

In order for MNMs to perform pollutant removal tasks in specific environments, it is necessary to integrate various functional modules into the MNMs in order to ensure their functionality. The structural materials are responsible for providing robust support properties, the actuator module ensures the proper functioning of the autonomous movement, and the surface materials interact with the surrounding environment (including the interaction between the MNMs and the environmental pollutants) due to their intrinsic properties or the presence of functional groups on the surface. The integration of functional modules is a complex multi-step process, which can be broadly classified into two main categories: top-down and bottom-up. Top-down strategies include physical vapor deposition, 3D printing and roll-up, lithography, and reactive ion etching, while bottom-up strategies encompass electrodeposition and self-assembly. Although the top-down design scheme is more complex and time-consuming for the construction, synthesis, and fabrication of MNMs, it is more conducive to the achievement of precisely controlled and high-performance MNMs. The bottom-up strategy, on the other hand, is promising for industrial-scale production due to its simplicity and scalability. Both strategies have distinctive advantages. Further investigation into the complementary integration of these synthesis techniques and the appropriate optimization of solutions would be advantageous in achieving MNMs’ enhanced performance, stable operation, and adaptability to complex environments.

### 2.1. Propulsion Mechanism

#### 2.1.1. Fuel-Driven MNMs

MNMs can be categorized based on their motion mechanisms, which differentiate between fuel-driven and external-field-driven types [73] (Figure 3). Fuel-driven MNMs utilize a chemical fuel or substrate to generate movement through a reactive process [74,75,76,77]. Although H_2_O_2_ is known for its toxicity, it is still the most widely employed fuel in these applications. Pt, despite being costly, serves as the most efficient catalyst for the decomposition of H_2_O_2_. The propulsion of MNMs is facilitated through bubble jetting or self-phoresis [78]. The latter refers to the spontaneous movement of particles induced by a gradient, which can involve factors such as solute concentration (self-diffusiophoresis) or electric potential (self-electrophoresis). For instance, the self-propulsion of Pt MNMs in H_2_O_2_ is caused by the continuous jet stream generated by O_2_ bubbles [79,80], which also enhances thrust and mobility in high ionic strength environments, such as seawater (Figure 3a). Enzymes like catalase, urease, and glucose oxidase serve as biological equivalents to inorganic catalysts [81,82,83]. The uneven distribution of MNMs within catalase promotes movement in H_2_O_2_ via a bubble propulsion mechanism. In contrast, urease and glucose oxidase are able to facilitate catalytic processes with more biocompatible substrates, particularly urea and glucose. This enzymatic action creates a gradient of products, which facilitates the self-phoresis of MNMs by generating a localized force that propels their motion (Figure 3b). Despite this promising mechanism, the use of enzymatic systems is largely constrained to biomedical applications, as these systems rely on the presence of naturally occurring substrates within specific environments [84]. To overcome this limitation, researchers have sought alternative catalytic approaches and fuel sources that can extend the functionality of MNMs to non-biological contexts. Such efforts aim to adapt MNMs for applications in areas like environmental remediation and industrial manufacturing, where natural substrates are either scarce or absent. A key example of these alternative mechanisms is the catalytic breakdown of MNMs driven by chemical reactions that consume both the MNMs and the fuel, as seen in systems like Mg-H_2_O [85,86,87] (Figure 3c). Nonetheless, fuel-driven MNMs face limitations in terms of motion control, including restricted on/off switching and directionality, as well as a shorter operational lifespan.

Chemically propelled MNMs primarily utilize chemical energy from fuel sources to achieve autonomous motion. Commonly used fuels include H_2_O_2_, Br_2_, I_2_ solutions, hydrazine, as well as acidic and alkaline solutions. In addition to utilizing the chemical energy carried by the fuel, MNMs can achieve motion based on ion exchange in response to cations (Ca^2+^/Na^+^, Pb^2+^) in the salt solution [88,89]. Based on ion exchange, ions can not only trigger MNMs to achieve a gel-solution phase transition [88] but also induce morphological changes (swelling/shrinking) in MNMs through ion chelation [89]. The dynamic properties of ionic equilibrium and ion-responsive behavior can be regulated by ion power and degree of ionization. Chemically driven MNMs offer advantages such as ease of application and a wide range of propulsion reactions. However, these chemical propulsion systems often lack effective control over the kinematic properties of MNMs, necessitating the use of external fields for precise movement regulation. Moreover, the operational lifespan of chemically propelled MNMs is typically limited.

#### 2.1.2. Externally Driven MNMs

External field-driven MNMs overcome the limitations present in fuel-driven counterparts by tapping into energy from various external sources, such as sound, light, temperature, and electromagnetic fields [90,91]. Among these options, light stands out as a natural, abundant, and renewable energy source capable of initiating multiple degradation mechanisms at once. Consequently, it is clear that employing light as a propulsion method for MNMs offers substantial advantages [92,93,94,95,96]. The movement of photodynamic MNMs arises from the creation of electron–hole pairs (e^−^-h^+^) in photoactive materials upon irradiation. For instance, when a semiconductor is exposed to photon energy that surpasses its bandgap, e^−^ are excited into the conduction band, generating corresponding holes in the valence band. These photogenerated carriers subsequently catalyze additional photochemical reactions, enabling motion via mechanisms such as bubble propulsion or self-phoresis. To inhibit the recombination of light-generated carriers within the semiconductor and to preserve the symmetry of the semiconductor material, it is frequently essential to integrate metal with semiconductor substances, thereby forming a metal–semiconductor junction [97,98]. This junction plays a crucial role in preventing carrier recombination while sustaining the symmetry of the semiconductor structure. Typically, Janus MNMs made from metal–semiconductor configurations demonstrate a motion mechanism where photogenerated carriers promote oxidation and reduction reactions of H_2_O on both sides of the MNMs (Figure 3d). Specifically, the semiconductor element acts as a source of H^+^ ions, which are subsequently utilized on the metal side, functioning as an H^+^ sink. This process establishes a charge gradient that generates a local electric field, which in turn propels the motion of MNMs through self-electrophoresis. However, even metal-free coated MNMs with a single component can achieve movement under light due to their intrinsic asymmetry or asymmetric illumination [99]. The main limitations faced by photodynamic MNMs include the attenuation of light intensity with depth and the decreased effectiveness of self-electrophoresis in environments with high ionic strength, such as seawater. In conclusion, light radiation induces asymmetric photocatalytic reactions on different surfaces of MNMs, creating charge gradients that facilitate autonomous movement via the self-electrophoresis mechanism [99]. However, the significant decrease in light intensity with increasing depth, along with reduced self-electrophoresis efficiency in high-ionic-strength environments, can limit the mobility of MNMs.

Magnetic fields offer a means to effectively control MNMs, providing exceptional maneuverability across various mediums [100,101]. Instead of relying solely on the principles of attraction and repulsion mediated by magnetic forces, MNMs can navigate through magnetophoresis when subjected to magnetic field gradients or via torque transfer in rotating magnetic environments. For example, helical MNMs can transform rotational movement into linear motion within a rotating magnetic field (Figure 3e) [102]. Unlike MNMs driven by optical methods, those manipulated magnetically do not possess autonomy and face significant limitations in terms of cost and size due to the requirements of magnetic setups [103]. The incorporation of magnetic materials such as Fe, Co, and Ni as the magnetic layer of MNMs facilitates their movement in various environmental media when subjected to a magnetic field. This characteristic enables remote control and recyclability of the MNMs [101]. However, the unique shapes and operational mechanisms of magnetic MNMs are often limited by complex fabrication processes and demanding configurations [103].

Propulsion via electric fields occurs by integrating materials that have different polarization rates, similar to the mechanisms used in magnetic propulsion [104]. Electric field-driven MNMs offer notable advantages, including high precision and rapid speed regulation. The combination of direct current (DC) and alternating current (AC) electric fields facilitates the transport, activation, and deactivation of MNMs along predefined paths, with real-time speed adjustment to meet operational requirements [103]. The DC electric field primarily controls the transport speed of MNMs through electrophoresis and electroosmosis, while the AC electric field independently manages their alignment and orientation by exerting electric torque on the induced dipole of the MNMs. However, in practical applications, MNMs must navigate autonomously in complex fluid environments, where non-uniform electric field distribution and variations in medium conductivity can pose challenges. Additionally, acoustic fields can effectively drive MNMs of various structures, such as bimetallic nanowires and asymmetric nanowires, by applying pressure from ultrasound radiation (Figure 3f) [105,106,107,108]. The use of ultrasound facilitates rapid internalization within cells, enhances intracellular movement, and allows for the precise delivery of therapeutic agents without damaging human tissues, making it particularly valuable in biomedical applications [109,110]. In order to facilitate acoustic manipulation, ultrasonically propelled MNMs frequently require the adoption of bespoke geometries or the incorporation of vibrating units to assist in their motion. Nevertheless, the applications of ultrasound-propelled MNMs remain constrained by the necessity for sophisticated apparatuses to generate ultrasound waves and the prerequisite of particular circumstances for the formation of standing waves.

In order to enhance the application of MNMs in living organisms, the prevailing approach nowadays is to combine particles with biological units to construct biohybrid MNMs. The majority of biological MNMs are composed of biodegradable hydrogels [111], biological cells [112], microorganisms [113], and biomolecules [114]. The components of bio-based MNMs are highly biocompatible and biodegradable, which enables MNMs to cross the blood–brain barrier and evade immune system tracking, thereby enhancing their application in living organisms [113,114]. The movement of bio-based MNMs is primarily regulated by enzyme-induced bond breaking and the morphological alterations of physically/chemically modified biomolecules, biological tissues, and biological cells.

In conclusion, external field-driven MNMs have the potential to meet the demands of various applications. However, further research is needed to develop optimal strategies for the hybrid integration of different external energy fields, advancements in material innovation, and improvements in single driving mechanisms. These efforts will enhance the pollutant removal efficiency of MNMs and broaden their applicability in environmental remediation.

### 2.2. Motion Control

The movement of MNMs can be induced by chemical fuels or external fields, with each approach presenting distinct advantages and drawbacks (Table 2). In real-world applications, effectively controlling MNM motion is essential to satisfy the specific demands of various tasks. In fluid environments, the motion of MNMs primarily manifests as irregular Brownian motion, where both direction and velocity play critical roles in determining their performance. Velocity regulation is mainly accomplished through chemical reactions, which can significantly improve transport and transmission efficiency. Additionally, controlling direction, primarily achieved via the application of magnetic fields, enhances the accuracy and precision of MNMs. In current applications, external drives are extensively employed to regulate the motion behavior of MNMs, thereby addressing issues such as secondary contamination, limited controllability, and the short operational lifespan inherent in chemically driven systems. Magnetic fields, electric fields, and light sources are applied to the magnetic components, conductive surfaces, longitudinal axes, and sides of MNMs, respectively, to produce asymmetric charge distribution, torque, force imbalance, and photocatalytic reactions. These mechanisms enable effective control of both the speed and direction of MNMs.

#### 2.2.1. Speed Control

Fuel serves as a vital source of chemical propulsion, and the regulation of velocity for chemically driven MNMs is accomplished by adjusting the energy input, specifically the concentration of the fuel. Generally, the speed at which MNMs move exhibits a direct positive correlation with fuel concentration; in other words, as the concentration of fuel increases, the reaction rate also escalates.

In their research, Mathesh et al. utilized a simple methodology involving soft nanostructures to create enzyme-driven two-dimensional (2D) MNMs that can move autonomously on an exceptionally low concentration of fuel (0.003% H_2_O_2_). As the concentration of fuel rose from 1 mM to 5 mM, the motility of the 2D MNMs increased from 3.5 ± 0.04 µm/s to 6.53 ± 0.68 µm/s. Due to the buoyancy of solutes, the 2D MNMs exhibited effective positive chemotaxis and demonstrated resilience against gravitational influences during movement. Additionally, these 2D MNMs were capable of swiftly and effectively removing methylene blue dye, achieving a removal efficiency of up to 85% [120]. Chemically propelled MNMs not only have excellent catalytic properties and autonomous propulsion capabilities but also accelerate diffusion-limited processes, making them uniquely promising for environmental remediation applications. However, due to energy loss during the conversion process, the resulting speed of movement is often suboptimal. For example, in the self-electrophoresis propulsion mechanism, energy loss during the conversion in MNMs occurs in four stages (Figure 4a): the first stage is due to the chemical and non-chemical decomposition of fuels (H_2_O_2_) by catalysts (Pt) [121]; the second stage results from the use of highly exergonic reactions to generate a potential drop along the MNMs’ surface [122]; the third stage of energy loss is caused by the limited potential difference between the cathode and anode, which restricts the electrochemical energy converted into mechanical energy [123]; and the fourth stage involves energy loss due to electroosmotic flow, which opposes the self-electrophoretic motion [124,125]. Similarly, the inefficiency of the bubble recoil mechanism in propelling MNMs can be understood in two stages of energy loss (Figure 4b). First, the O_2_ produced must be expanded into sufficiently large bubbles to detach from the MNMs’ surface, but only a small fraction of the energy (1%) is used for bubble expansion, with the remainder dissipated as heat. Second, at low Reynolds numbers, propulsion occurs only at the instant of bubble release, with acceleration and deceleration times limited to microseconds, so the recoil motion lasts only a brief moment [126]. Therefore, in order to enhance the pollutant removal efficiency of MNMs, it is necessary to use external regulation to enhance the movement speed of MNMs. For example, Huang et al. developed a straightforward and adaptable approach to effectively assemble MOF NPs into durable structures known as MOFtors. By utilizing the transient Pickering emulsion method, Fe_3_O_4_@NH_2_-UiO-66(Fe-UiO) nanoparticles were rapidly self-assembled into large-scale colloidosomes referred to as Fe_3_O_4_@NH_2_-UiO-66 colloidosomes (FeUiOsomes). The Fe-UiOsomes-Pt MNMs displayed remarkable motility in a 5 wt% H_2_O_2_ solution, as indicated by bubble formation during the chemical reduction process. The removal efficiencies achieved for Cr^6+^ and methyl orange were 91% and 94%, respectively [127].

In addition to tuning the rate of motion of MNMs by adjusting fuel concentration, pioneering research by Josep Wang’s group has demonstrated that thermal control is another viable technological tool [128,129]. An increase in temperature significantly enhances the propulsion speed of Pt/Au nanowires. Similarly, a comparable effect is observed in bubble-propelled microtubules, which are commonly utilized to mitigate the influence of reduced fuel availability on propulsion efficiency. The solution temperature can be regulated using two Peltier elements connected to a direct current (DC) power supply, positioned beneath the sample containing the microtubules. Heating the system to physiological temperatures not only enhances the efficiency of microtubule propulsion but also enables movement in low concentrations of H_2_O_2_ fuel [130]. Additionally, soft MNMs composed of flexible thermo-responsive polymer microjets can adapt to temperature changes in their surrounding solution. These MNMs can reversibly fold and unfold with high precision, enabling multiple cycles of activation and deactivation as their shape transitions alter their radius of curvature. The incorporation of stimuli-responsive materials represents a promising strategy for the future development of intelligent MNMs. Moreover, stimulus-responsive behaviors such as structural expansion/contraction/disorder/bending induced by photothermal effects can also be used to modulate the speed of movement of MNMs [131,132,133,134,135].

#### 2.2.2. Directional Control

The small size of MNMs makes them vulnerable to Brownian motion in complex fluid environments, often causing a mismatch between their intended and actual movement. To counter this effect and enable precise directional control, magnetic materials are typically embedded within the MNMs, allowing them to transport loads accurately under the influence of an external magnetic field.

Feng et al. developed non-invasive Ni/Zn MNMs driven by H_2_O, where H^+^ is transformed into Zn^2^^+^ and H_2_ at the Zn end. The motion of these MNMs arises from the combined influence of a concentration gradient and an internally generated electric field caused by the progressive buildup of Zn^2^^+^. The Ni end enables precise targeting of cells, offering significant advantages in controlling MNMs’ movement [136]. After producing tubular Ti/Fe/Pt MNMs and examining their controlled movement within a microfluidic channel, Sanchez et al. found that the MNMs’ directional motion, aligned with the channel’s flow, was influenced by water movement. They also demonstrated that the MNMs could adjust their velocity using a magnetic field, facilitating precise delivery of the product to a specified target [137].

### 2.3. Multifunctionality

To perform specialized functions within specific applications, MNMs must not only demonstrate autonomous movement but also be capable of carrying out tailored tasks. Thus, incorporating a variety of modules—both organic and inorganic components—is crucial for enhancing their adaptability and functionality [138,139,140,141]. Each module within MNMs serves a distinct purpose: structural components ensure a stable framework, engines generate motion, imaging materials facilitate tracking, magnetic elements introduce magnetism for retrieval via magnetic fields, and surface components interact with their environment through inherent properties or functionalized surface groups. For instance, DNA-based MNMs leverage self-propulsion, programmability, and the precision of Watson–Crick base pairing to facilitate intracellular cancer biomarker detection or gene delivery [142]. The integration of various components brings several challenges, particularly in fabrication, assembly, and scalability. Consequently, thorough assessment of MNMs for both single-function or multi-pathway applications is crucial prior to their implementation in targeted environments [143].

Although the positive results noted in controlled laboratory experiments are promising, real-world environments present complex and continually changing conditions. The propulsive and locomotor functions of MNMs can be affected by variations in the chemical makeup of the surrounding medium or by exposure to unexpected stimuli. Therefore, similar to living organisms, intelligent MNMs must exhibit the ability to adjust their behaviors in response to the dynamic shifts that characterize their surroundings. For instance, E. coli can detect the concentration of chemicals (nutrients) in its vicinity using specialized receptors. Utilizing this sensory information, the bacterium can determine whether it should move towards areas with higher chemical concentrations [144]. In addition, the vertical migration of plankton within aquatic ecosystems in response to daily fluctuations exemplifies another form of adaptive behavior. Microorganisms exhibit diel vertical migration, a behavioral pattern that enables them to optimize survival by ascending to the surface at night to forage for food and evade predators, then retreating to deeper waters during daylight hours to minimize exposure to threats [145]. Inspired by such adaptive strategies in nature, MNMs have been designed to navigate their environments through mechanisms such as chemotaxis, phototaxis, and magnetotaxis. These movements are governed by external stimuli, showcasing their ability to respond dynamically to environmental cues [146,147]. For instance, chemotactic MNMs are known to navigate according to the concentration gradient of fuels or surfactants [148,149,150]. As MNMs align with this gradient, their diffusion rate significantly increases due to the heightened catalytic reaction rates observed at elevated fuel concentrations (Figure 5a). Moreover, phototropic MNMs possess the ability to detect light direction, allowing them to orient themselves and move toward or away from light sources (Figure 5b) [151,152,153,154]. Phototaxis is often clearly observable in highly asymmetric structures, and this phenomenon has been demonstrated by the pioneering research of Guan’s group, where smart photo-isotropic semiconductor MNMs were generated by using the finite penetration depth of light to induce chemical reactions on the asymmetric surface of MNMs [155]. Subsequently, another parallel study of MNMs propelled by phototaxis was reported by Dai’s team. In this, Janus MNMs were composed of Si trunks and TiO_2_ branches. These structures not only demonstrated programmable phototropic responses, either negative or positive, but also exhibited behavioral traits similar to those found in natural green algae [156]. Another distinct behavioral response is gravitaxis, characterized by movement influenced by gravitational forces. This response has been noted in MNMs that possess an asymmetric mass distribution or structural arrangement, resulting in movement that opposes gravity. For instance, certain photopelled MNMs have shown a gravitactic response to light, illustrated by the previously mentioned diurnal vertical migration, commonly termed “negative photogravitaxis” (Figure 5c) [157,158,159]. Additionally, MNMs with magnetic responsiveness can alter their movement based on the gradient of the magnetic field or the direction of a dynamic magnetic field, thereby optimizing energy usage [160,161,162]. For example, when the polarity of the magnetic field is reversed, it can apply either an attractive or repulsive force on the MNMs (Figure 5d).

### 2.4. Swarm Behavior

The term “group behavior” refers to the collective execution of tasks that surpass the abilities of individual organisms. For instance, ant colonies can self-assemble to form resilient structures, like bridges, allowing them to navigate gaps in the environment. Similarly, starlings perform synchronized movements, mimicking cloud-like formations, to avoid predators. This coordinated movement in natural organisms serves as a foundational model for developing group movement strategies in artificial MNMs [163]. Compared to solitary MNMs, group MNMs exhibit greater efficiency, robustness, and flexibility. For example, a collective of MNMs can complete tasks more quickly and carry out size-dependent operations more efficiently than a single MNM. Additionally, large loads are moved by groups of MNMs, a task typically requiring the cooperation of multiple MNMs. Group MNMs also possess the ability to detect a wider range of environmental changes and maintain functionality even when one member malfunctions [164]. The key characteristic of an MNM population is the collective movement of its individual components. Strategic MNMs generally move in a coordinated manner in response to energy gradients [165]. However, in real-world scenarios, groups of MNMs must adapt to changing environments through self-organization or by altering their shape and function. These adaptive behaviors rely on physicochemical interactions between MNMs [166,167,168,169,170]. For instance, photopropelled MNMs can spontaneously assemble into “serpentine” microchains via shape-modulated magnetic dipole–dipole interactions [171]. Additionally, when exposed to an external magnetic field, MNMs can transition reversibly from a dispersed to an aggregated state, allowing them to explore their surroundings more effectively. At the same time, MNMs, which achieve movement based on ion exchange, not only exhibit fascinating swarming behaviors in fluid environments but also possess the ability to pump nearby material towards the collective moving structure. The pumping activity of MNMs leads to the growth of moving clusters, and their speed of movement increases dramatically with increasing volume [117]. Moreover, in low Reynolds number environments, these self-assembled dynamic populations also can perform stable linear motions [172]. Such dynamic structures are capable of trapping nearby microscopic matter while purifying the liquid used as a transport medium and fuel. This interesting phenomenon opens the door to potential applications in the field of water remediation that are currently being developed. Such MNMs are capable of overcoming obstacles, moving, rotating, and even transporting loads collectively [173]. Thus, integrating pollutant removal and degradation functions within an MNM population presents a promising approach to addressing environmental remediation challenges.

Effective communication between MNMs or with their environment is essential for achieving synchronized group movements, yet it poses a significant challenge. For instance, MNMs can exchange chemical signals, such as releasing small molecules or ions, which create chemical gradients that attract, repel, speed up, or slow down other MNMs [174]. Additionally, telehydrodynamic communication leads to a “hit and run” effect, where one MNM acts as a “leader”, gathers smaller “followers”, and attempts to secure the support of competing “followers” [175]. This intriguing interaction can be utilized to design MNMs capable of detecting and neutralizing more persistent and harmful pollutants in water with greater precision. The gathered information is then shared with the rest of the MNM population, prompting them to cluster around the pollutant to ensure its removal or degradation. However, in real-world applications, the exchange of such complex information and coordinated functions among MNMs remains a considerable challenge.

## 3. Environmental Remediation

Autonomous motion has been shown to enhance fluid mixing, increasing the likelihood of reactive oxygen species (ROS) interacting with pollutant molecules by a factor of 3.5 compared to non-autonomous motion. This effect promotes the oxidative degradation of pollutants under favorable conditions [176]. However, in open environments like ocean currents, MNMs’ movement may be disrupted, hindering their recycling and reuse. As a result, environmental remediation with MNMs is generally performed in controlled, confined settings. Additionally, applied external fields can be used to guide MNMs within confined environments, reducing the potential for secondary contamination (Figure 6). MNMs are particularly effective for degrading pollutants at the micro- and nanoscale, and their application in environmental cleanup involves both detecting and removing contaminants [177,178,179,180,181,182].

### 3.1. MNM-Based Environmental Remediation Systems

Although MNMs are not ideal for remediation in open waters, they can effectively remove or degrade pollutants in confined environments, such as enclosed vessels used in offshore operations. A small-scale environmental remediation system utilizing MNMs should include a dedicated tank with an inlet for polluted water, an outlet for clean water, and sensors to monitor water quality in real time. These tanks can be exposed to sunlight or fitted with UV lamps to provide a continuous energy source for MNMs’ photopropulsion. To enhance the efficiency of pollutant degradation, increasing the MNMs’ speed may be necessary, which can be achieved using H_2_O_2_ as fuel. However, magnetic propulsion may be more effective than optical drive, especially in highly conductive brackish waters. High conductivity diminishes the self-generated electric field required for the motion of most photopropelled MNMs, thus limiting their performance. Surrounding the water tank with orthogonal pairs of electromagnetic coils, controlled externally, allows the operator to guide the MNMs. This system accelerates pollutant adsorption or degradation and aids in collecting MNMs at a specified location during the treatment, ensuring the release of purified water. Additionally, magnetic actuation can direct MNMs to a secondary container, where non-degradable contaminants can be safely released, allowing MNMs to be reused.

To meet the technical demands described, an ideal MNM must consist of multiple components, each serving specific functions. For example, incorporating a magnetic engine as the core component allows for precise control over its position, speed, and movement path by adjusting the magnetic field settings. To enhance the adsorption efficiency, this engine can be surrounded by a highly porous material with a large surface area that acts as an adsorbent. The adsorbent layer may then be encased within a shell of photoresponsive materials capable of initiating sequential photodegradation pathways. To further improve degradation efficiency, the MNM’s surface can be functionalized with enzymes to accelerate contaminant breakdown [183]. However, it is essential to reassess the enzyme’s stability when exposed to light or light-induced ROS to ensure that the system remains efficient throughout the remediation process. Such systems must also be environmentally sustainable and more cost-effective than traditional contaminant treatment methods. A full economic evaluation of these MNM-based systems should account for energy usage related to pumping, water quality sensors, controllers for magnetic fields, light sources, and the critical factor of treatment duration. The treatment efficiency depends on variables such as pollutant type, MNM quantity, magnetic field parameters, and light intensity.

### 3.2. MNMs for Environmental Sensing and Monitoring

The use of self-propelled MNMs introduces a novel approach for real-time environmental monitoring, offering substantial potential in detecting sudden changes and emerging threats. These systems also show promise in continuously tracking remediation processes or monitoring hard-to-reach areas with minimal sample requirements. One of the earliest strategies for using MNMs in environmental remediation came from the observation that their motor behavior changes in response to specific chemical hazards [184,185]. Catalytic MNMs, for example, display chemotactic movement by moving towards regions with higher H_2_O_2_ concentrations, with an accompanying increase in velocity due to the concentration gradient [186]. Polymer microspheres embedded with Pd nanoparticles have been observed to converge towards areas of higher pH levels, and the rate of movement rises with increasing pH. This suggests MNMs are well-suited for monitoring environmental pH changes [187]. In summary, MNMs can be utilized to identify the origin of chemical plumes by implementing a convergent search mechanism. For example, bimetallic nanowire MNMs exhibit a significant and selective speed increase when exposed to Ag^+^ ions. This accelerated motion is attributed to the underpotential deposition of Ag in the Au/Pt segmentation, which results in enhanced motion of the MNMs due to the greater deposition of Ag on the Au side. This movement-based sensing technique offers quantitative data on contaminants by analyzing the changes in the MNMs’ speed or displacement and is highly selective, with sensitivity down to nanomolar concentrations [49,188]. The Ag-enhanced acceleration provides a strong platform for motion-based DNA hybridization detection, which involves Ag nanoparticle labeling [189]. The speed of Ir-based Janus MNMs, which are powered by low concentrations of hydrazine, exhibits a strong dependency on the fuel concentration. This characteristic suggests their potential utility in monitoring hydrazine plumes [190,191]. Likewise, tubular MNMs driven by catalase show a significant decrease in both motility and lifespan when subjected to heavy metals, pesticides, nerve agents [192,193], and HCl and NH_3_ gases [194]. This approach to toxicity screening capitalizes on the inhibitory impact of toxins on catalase, a key enzyme that facilitates MNMs’ movement [193]. Similarly, the application of enzyme-free MNMs for environmental detection is founded on the notion that specific pollutants can poison the Pt catalytic layer, leading to a decrease in their movement rate. For instance, Pumera utilized tubular MNMs made of Cu/Pt to selectively detect Pb^2+^ while differentiating it from Cd^2+^, a selectivity resulting from the differing adsorption rates of these metals on the Pt catalytic layer [195]. A comparable methodology has been employed to identify dimethyl sulfoxide and thiol-containing amino acids or peptides, which also hinder the activity of the Pt catalyst [196].

The integration of nanoparticles, including fluorescent dyes and quantum dots (QDs), into MNMs presents a compelling strategy for creating advanced environmental microsensors. Upon interacting with pollutants, the fluorescence intensity of these MNMs experiences a swift burst and subsequent recovery within a brief time interval. This cutting-edge microsensor facilitates the visualization of environmental monitoring activities, offering a more intuitive approach to such assessments. Figure 7A depicts an example of QDs embedded within tubular MNMs designed for heavy metal detection. These tubular MNMs were synthesized by attaching CdTe QDs to the external surface of PEDOT/Pt-based MNMs, using a positively charged layer of poly(diallyldimethylammonium chloride) to secure both the inner catalytic and outer sensing layers. The enhanced affinity of trace Hg^2^^+^ ions for QDs leads to a significant fluorescence emission burst, enabling effective differentiation between the two prevalent mercury species, Hg^2^^+^ and CH_3_Hg^+^, as well as other competing ions in real time [197]. The high toxicity of Cd^2+^ poses substantial challenges in water treatment processes. Tian et al. introduced an innovative, simple, and cost-effective microsensor based on N and P co-doped carbon dots (N, P-CDs), synthesized via a one-step hydrothermal method, capable of detecting Cd^2+^ and ascorbic acid with high sensitivity and selectivity [198]. The N, P-CDs were shown to detect Cd^2+^ through mechanisms involving the inner filter effect and static quenching, with the fluorescence quenching efficiency demonstrating a strong linear correlation with Cd^2+^ concentration. Additionally, the N, P-CDs/Cd^2+^ hybrids functioned as “turn-on” fluorescent sensors for ascorbic acid detection. Notably, these sensors maintained a high recovery rate after detecting heavy metal ions in water, indicating significant potential for applications in food safety testing. However, further studies are necessary to evaluate the long-term stability and durability of these MNMs under various environmental conditions and to understand the influence of co-existing substances in water on their performance. Janus MNMs composed of Si/Pt and loaded with fluorescein amine have been employed for the “on–off” detection of diethyl chlorophosphate, which serves as a simulant for the chemical agents sarin and soman [199]. The fluorescence burst observed in these MNMs results from HCl, a by-product generated during the phosphoramidation reaction between diethyl chlorophosphate and fluorescein amine, leading to the disruption of fluorophore conjugation. Additionally, the functionalization of tubular graphene MNMs with specific ricin B aptamers and the incorporation of fluorescein amidine dyes present significant opportunities for the “on–off” fluorescence detection of various chemical reagents, including ricin [200]. Furthermore, molecularly imprinted tubular MNMs utilizing algal blue protein as the template molecule, alongside Pt and Ni as the catalytic and magnetic components, have been utilized to detect algal blue protein in aquatic samples [201]. The inherent fluorescence of this cyanobacteria-associated protein guarantees that water quality remains unaffected by pathogenic microorganisms.

The spread of infectious diseases through contaminated water and microbial pathogens poses a considerable threat to public health worldwide. Consequently, it is crucial to detect and isolate waterborne pathogens to fulfill the requirements of environmental monitoring. Research has shown that self-propelled functionalized MNMs are remarkably effective for the direct and selective extraction of target analytes from raw samples [202]. For instance, tubular MNMs based on Pt and modified with lectin facilitated the direct capture of E. coli from untreated seawater and drinking water samples (Figure 7B) [203]. Additionally, MNMs functionalized with an antibody derived from Bacillus globigii demonstrated the ability to selectively identify, capture, transport, and eliminate Bacillus globigii spores within the environmental context [204]. At the same time, the combined effects of Mg/Au Janus MNMs in electrochemical assessments using strip-based microvolumetric electrodes have been established. These MNMs serve dual roles, acting not only as “autonomous stirrers” that significantly improve mass transfer processes but also as “artificial enzymes” that facilitate the degradation of target analytes (Figure 7C). The aim is to catalyze the breakdown of these analytes. In their function as “artificial enzymes,” MNMs are localized on the sensor band’s surface, where they hydrolyze paraoxon into the easily detectable compound p-nitrophenol under alkaline conditions [205]. Likewise, Mg/Au Janus MNMs are effective in decomposing the non-electrically active persistent organic pollutant diphenyl phthalate (DPP) into phenol, which is also readily detectable [206]. Additionally, MNMs made from specially responsive materials and produced using screen-printing techniques can react to external stimuli, including light, pH, and temperature, resulting in noticeable color changes. This feature shows considerable potential for efficient, visual “real-time” monitoring of water quality [207].
Figure 7(**A**) Detection of pollutants through motion-based methods employing self-propelled MNMs. (**a**) Biocatalytic MNMs made from PEDOT-Au-catalase that are designed for screening heavy metal ions. This method demonstrates how pollutants influence the speed of MNMs by inhibiting the activity of the catalase biocatalytic layer. Reproduced from Ref. [193]. Copyright 2013, the American Chemical Society. (**b**) MNMs modified with CdTe quantum dots (QDs) are utilized for the selective detection of mercury, relying on the fluorescence quenching of the QDs that occurs upon interaction with Hg and which is immobilized on the MNMs’ surface. Reproduced from Ref. [197]. Copyright 2015, the Royal Society of Chemistry. (**B**) PEDOT/Pt MNMs modified with lectins enable the selective capture, transport, and release of Escherichia coli bacteria from environmental samples. Reproduced from Ref. [203]. Copyright 2012, the American Chemical Society. (**C**) The detection of non-electroactive nerve agents in sensing strips is facilitated by Mg/Au Janus MNMs. The localized generation of hydroxide ions by these MNMs, combined with convection forces, leads to the breakdown of organophosphate (OP) nerve agents into easily detectable nitrophenol. Reproduced from Ref. [205]. Copyright 2015, the Royal Society of Chemistry. (**D**,**E**) MNMs for pollutant detection and degradation. (**D**) Three-dimensional hierarchical HRP-MIL-100(Fe)@TiO_2_@Fe_3_O_4_ Janus magnetic MNMs as a smart active platform for detection and degradation of hydroquinone. Reproduced from Ref. [138]. Copyright 2022, the American Chemical Society. (**E**) A 3D hierarchical LDH-based Janus micro-actuator for detection and degradation of catechol. Reproduced from Ref. [208]. Copyright 2022, the Elsevier B.V. All rights reserved.
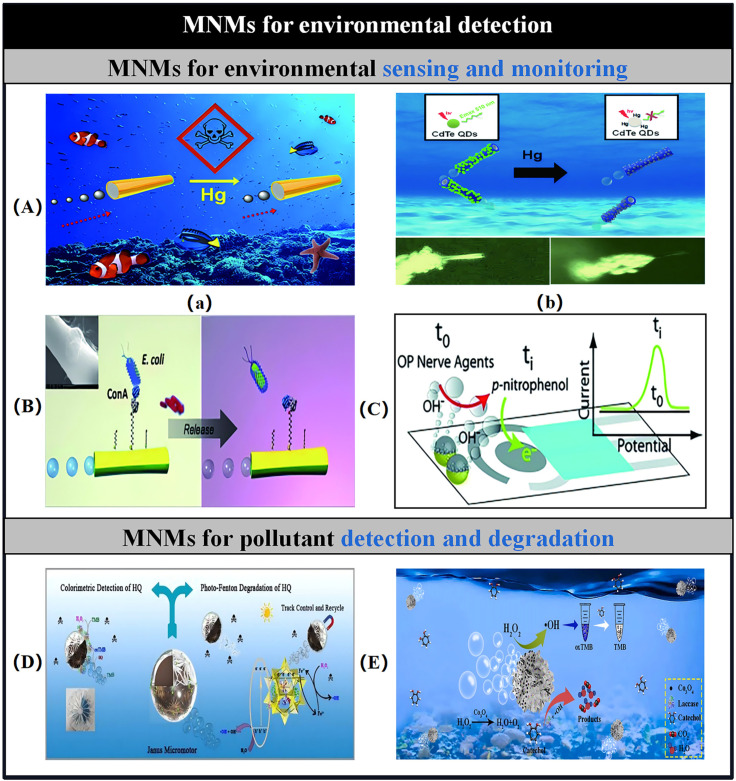


Detection of specific environmental pollutants can be achieved by leveraging changes in the kinematic behavior of MNMs in response to pollutants or utilizing the natural fluorescence properties of synthetic materials. However, in the field of pollutant detection and removal, traditional MNMs that serve solely as detection tools face significant limitations. Despite their high detection efficiency and sensitivity, such MNMs often lack pollutant removal capabilities, necessitating additional removal steps post-detection, which imposes extra time and resource burdens. Additionally, single-function MNMs are susceptible to interference from various factors in complex environments, which diminishes detection accuracy and stability.

Consequently, researchers have started exploring the development of multifunctional MNMs that integrate both detection and degradation capabilities [138,208,209]. For example, Yang et al. developed horseradish peroxidase (HRP)-modified, three-dimensional layered MIL-100(Fe)@TiO_2_@Fe_3_O_4_ Janus magnetic MNMs. The combination of autonomous propulsion, high peroxidase-like activity, a tuned heterojunction with matched energy band structure, and a 3D hierarchical structure enabled dynamic, sensitive detection and photo-Fenton degradation of hydroquinone (HQ). Furthermore, these MNMs exhibit a wide pH adaptation range and can be magnetically recycled after completing degradation tasks (Figure 7D) [138]. Similarly, Xing et al. presented the first proof-of-concept for ascorbic acid-functionalized NiMn-CLDHs@HNTs-Ag self-propelled MNMs. These novel MNMs facilitated sensitive, low-detection-limit colorimetric phenol detection and provided sustained propulsion through the catalytic decomposition of H_2_O_2_ by Ag particles, showcasing how highly active sites in the 3D hierarchical structure enhance reaction efficiency [209]. A follow-up study demonstrated Janus MNMs made of laccase-functionalized MgAl-CLDHs and Co_3_O_4_-C nanoparticles, which, due to their autonomous motility and high peroxidase activity, achieved sensitive catechol detection and degradation at lower fuel concentrations (Figure 7E) [208]. Integrating detection and degradation functions into a single MNMs markedly improves environmental remediation efficiency, particularly in complex environments that require high selectivity and stability. Future research should focus on evaluating the performance of multifunctional MNMs in multi-component mixtures to ensure long-term stability and sustainable application across various water quality conditions.

### 3.3. MNMs for Adsorption Removal of Pollutants

The concept of “adsorption” refers to the mechanism through which atoms, ions, or molecules from gases, liquids, or dissolved solids are drawn to the surfaces of MNMs. A primary focus of researchers in the field of MNMs for environmental remediation has been the development of MNMs functionalized with suitable biological ligands or constructed from materials with a high surface area, such as graphene or activated carbon. Utilizing MNMs for dynamic degreasing offers an effective solution for addressing oil spills, garnering significant interest within the research community. For instance, Guix et al. [50] employed tubular MNMs made from alkanethiol-modified PEDOT/Pt, exploiting the adhesion and permeation properties of the alkanethiol coatings to effectively purify oil-contaminated water (Figure 8A). Careful optimization was performed to ascertain the ideal length of the alkanethiol chains in relation to the sailing duration, allowing for the efficient simultaneous capture and transportation of multiple oil droplets. Additionally, these MNMs can be reused after the oil droplets they capture have been released and degraded, thus providing an innovative approach to the recycling of MNMs in practical applications. Following this, MnFe_2_O_4_@oleic acid pot-like bubble-propelled MNMs with hydrophobic characteristics were employed to tackle environmental oil contamination. Due to the presence of long oleic acid chains that impart a hydrophobic surface on the MNMs, there was no necessity for further surface modifications with additional materials [210]. Tubular MMNs, created using a wax-printed membrane-assisted convolution technique and consisting of graphene/Pt, have proven effective in extracting oil droplets from water samples. The combination of graphene’s high surface area-to-volume ratio and the kinematic properties of these self-propelled MNMs creates optimal conditions for hydrophobic collection of oil droplets from aqueous environments [211]. To avoid the use of H_2_O_2_ fuel for propulsion, Pumera et al. [12] employed MNMs made from dodecyl sulfate/polysulfone polymers, which are millimeter-sized capsule-type MNMs driven by the Marangoni effect. These capsules repel oil droplets by interacting from distances of several centimeters and can merge smaller droplets, thus providing efficient cleaning of water surfaces. An alternative environmentally friendly strategy involves seawater-driven Au/Mg MNMs modified with self-assembled monolayers of long-chain alkanethiols [212]. In chloride-rich environments, Mg particles engage in a spontaneous redox reaction induced by galvanic coupling or pitting, enhancing the propulsive force and generating substantial amounts of H_2_ bubbles. Simultaneously, the strong hydrophobic properties of the modified gold layer facilitate the rapid collection of oil droplets, while the presence of a Ni layer aids in the effective recovery of Mg nuclei after dissolution. Additionally, the natural strong adsorption properties of commercial melamine sponges (MSs) present significant potential for oil–water separation. To facilitate autonomous movement on the water surface, Sun et al. designed MS MNMs with asymmetric multilayer structures modified by polydimethylsiloxane (PDMS) and molybdenum disulfide (MoS_2_) (Figure 8B) [213]. MoS_2_ served as a photothermal layer, providing propulsive force when exposed to near-infrared light. PDMS not only provided excellent superhydrophobic properties but also acted as an intermediate layer to enhance the adhesion between MSs and MoS_2_. This design enabled efficient autonomous motion, allowing the MNMs to effectively remove cyclohexane after eight cycles of use while maintaining stable photopropulsion performance across varying pH levels and in marine environments. The exceptional durability and recyclability of these MNMs provide new opportunities and sustainable strategies for directional propulsion in oil–water separation.

Catalytic MNMs derived from carbon isomers leverage the advantages of nanomaterials’ expansive surface area alongside the fluid mixing capabilities facilitated by MNMs’ movement, showcasing significant potential for the effective elimination of various pollutants [214,215]. For instance, self-propelling activated carbon MNMs were fabricated by asymmetrically applying Pt patches onto 80 μm activated carbon microspheres (Figure 8C). The substantial adsorption capacity of carbon adsorbents, combined with the swift movement characteristics of catalytic Janus MNMs, leads to the establishment of an efficient mobile platform for the adsorption of heavy metals, nitroaromatic explosives, organophosphorus nerve agents, and azo dyes [216]. Despite this, graphene remains the most commonly employed allotrope of carbon for pollutant capture and removal applications. Notably, Si/Pt Janus magnetic MNMs coated with reduced graphene oxide have shown an improved ability to remove persistent organic pollutants, such as polybrominated diphenyl ethers and triclosan, from water. These MNMs can be reused across four consecutive cycles while retaining their initial adsorption characteristics [217]. Furthermore, hybrid MNMs comprising ZrO_2_/graphene oxide/Pt have exhibited exceptional efficiency in capturing nerve agents within seconds. The external ZrO_2_ nanostructures in these hybrids are primarily responsible for their effective and selective binding to organophosphate compounds [218]. Advancing the engineering of graphene MNMs, Sanchez utilized graphene oxide-based MNMs to actively capture and eliminate Pb^2+^ and examined their reusability after chemically detaching Pb^2+^ from their surfaces [219]. However, the removal of heavy metal ions from aqueous solutions using adsorption-based methods is often limited by several factors, resulting in the low efficiency of conventional adsorbent-based MNMs constructions. First, sufficient contact between heavy metal ions and the adsorbent is necessary; second, the heavy metal ions must readily diffuse to and access both the surface and the interior of the adsorbent; and third, the adsorbent should possess an abundance of binding sites for heavy metal ions. To address these limitations, highly efficient MnFe_2_O_4_@MIL-53@UiO-66@MnO_2_ adsorbents were developed, incorporating self-propelled MNMs, distinct layered structures, and MOF composites (Figure 8D) [220]. The catalytic degradation of H_2_O_2_ by MnO_2_ enabled the adsorbent to achieve autonomous motion, thereby enhancing the contact probability between Pb^2+^/Cr^2+^ ions and the adsorbent. Additionally, the unique layered structure facilitates the diffusion and accessibility of Pb^2+^/Cr^2+^ ions to both the surface and interior of the adsorbent, while the abundant binding sites in the MOF complexes ensure stable binding of the heavy metal ions. Besides carbon isomers, zeolites are extensively used in MNMs synthesis due to their excellent capacity for retaining guest molecules in their microporous structures. Zeolite-based MNMs have been developed with dual functionalities: efficiently adsorbing neurotoxic agents and possessing antibacterial properties that can eliminate disease-causing bacteria in water [221]. These innovative materials represent a significant step forward in the development of advanced technologies aimed at mitigating various types of environmental pollution. While MNMs excel in addressing specific pollutants, the broader challenge lies in tackling persistent and pervasive contaminants that pose long-term risks to ecosystems. One such challenge is plastic pollution, which has become a critical global concern. Plastic materials contribute significantly to waste accumulation in marine environments and further break down into harmful microplastic particles. Leveraging the long-range electrophoretic attraction properties inherent in living organisms, self-propelled MNMs composed of recyclable ion-exchange resin spheres have been developed. These MNMs can dynamically remove and separate plastic particles from marine environments, achieving a sustainable removal efficiency of 90% [222]. The enzymatic degradation of plastic particles can be enhanced by incorporating active enzymes, such as lipase, on the surface of the MNMs [223]. These adhesive MNMs are anticipated to scavenge plastic particles from the marine environment through coordinated swarming behavior, introducing a novel aspect of MNMs’ functionality. However, further investigation is needed to understand the interactions between individual MNMs and between MNMs and the environment, as well as the applicability of these MNMs to various plastic compositions, sizes, shapes, and non-seawater samples.

Subsequently, researchers began initiatives aimed at environmental improvement by combining efficient biocatalytic agents with the enhanced motility properties of MNMs. For instance, Uygun et al. presented a highly effective mobile platform for CO_2_ scrubbing, addressing the issues associated with greenhouse gas buildup. This system integrates the biocatalytic functionality of carbonic anhydrase (CA) with the self-propelled movement of MNMs powered by H_2_O_2_ (Figure 8E). The CA enzymes attached to the MNMs surfaces facilitate the hydrolysis of CO_2_ into bicarbonate, which was then further mineralized to CaCO_3_. The ongoing motion of immobilized CA, coupled with improved mass transfer of the CO_2_ substrate, significantly boosts both the efficiency and speed of carbon sequestration compared to traditional stationary or free enzymes [224]. This dynamic property not only enhances the effectiveness of carbon trapping but also suggests that these mobile micro- and nanostructures have a wide range of potentials in environmental pollution treatment. For example, Hg^2+^ can be efficiently adsorbed by these microtubules due to its high specificity and strong binding ability to the mismatched pairing of T-T in DNA sequences, thus providing a novel approach for the treatment of heavy metal pollution [225]. Subsequently, multiple reports have outlined an environmentally friendly remediation approach that eliminates the need for H_2_O_2_ as a fuel source (Figure 8F). Mg/Au MNMs featuring dithiol (2-mercaptosuccinic acid) chelating layers have been utilized to form complexes with highly specific Zn^2+^, Cd^2+^, and Pb^2+^ ions. As the sample solution progresses, the Mg nuclei dissolve gradually, causing the MNMs to move while capturing heavy metal ions within the thiol layer. Subsequently, the gold shell rises to the surface along with the heavy metals, thereby facilitating the processing of the contaminants [226]. A notable example of an efficient biocatalytic remediation agent with enhanced motility properties is MNMs constructed using a metal–organic framework (MOF) as the primary structure [54,227]. Enzyme-powered porous MNMs built from MOFs, after undergoing ozonolysis, develop a sufficiently large hollow mesoporous structure, facilitating the adsorption and incorporation of peroxidase, thereby enabling autonomous motility. The hierarchical pore structure provides ample free space within the MNMs for the adsorption of contaminants, such as rhodamine B [54]. The integration of highly efficient biocatalytic agents with enhanced motility in MNMs significantly accelerates mass transfer and homogeneous mixing, thereby improving the contaminant degradation and removal process compared to conventional static methods. Additionally, incorporating thermosensitive materials into magnetic MNMs with adsorption capabilities allows for visualization of the pollutant treatment process [228]. For instance, pollutant treatment using thermosensitive magnetic MNMs composed of a pluronic tri-block copolymer (PTBC) enables the collection and disposal of toxic pollutants by modulating temperature. The temperature-dependent aggregation and separation of PTBC micelles can be monitored throughout this process. Furthermore, the contaminated MNMs exhibit significant self-cleaning capabilities that can be activated by temperature changes, maintaining high recycling retention even after multiple uses [229]. This combination of temperature-sensitive materials and magnetic propulsion introduces an innovative approach for water treatment and targeted contaminant removal using MNMs.
Figure 8(**A**) Dynamic oil extraction is facilitated by using SAM-modified Au/Ni/PEDOT/Pt tubular MNMs. Reproduced from Ref. [50]. Copyright 2012, the American Chemical Society. (**B**) Light-propelled super-hydrophobic sponge motor and its application in oil–water separation. Reproduced from Ref. [213]. Copyright 2023, the American Chemical Society. (**C**) Graphene/Pt/Ni tubular MNMs for Pb^2+^ removal. Reproduced from Ref. [219]. Copyright 2016, the American Chemical Society. (**D**) Self-propelled nanomotors based on hierarchical metal–organic framework composites for the removal of heavy metal ions. Reproduced from Ref. [220]. Copyright 2022, the Elsevier B.V. All rights reserved. (**E**) Self-propelled carbonic anhydrase PEDOT MNMs for CO_2_ sequestration. Reproduced from Ref. [224]. Copyright 2015, the Wiley-VCH. (**F**) Thiol-modified Mg/Au Janus MNMs for heavy metal removal. Reproduced from Ref. [226]. Copyright 2016, the Royal Society of Chemistry.
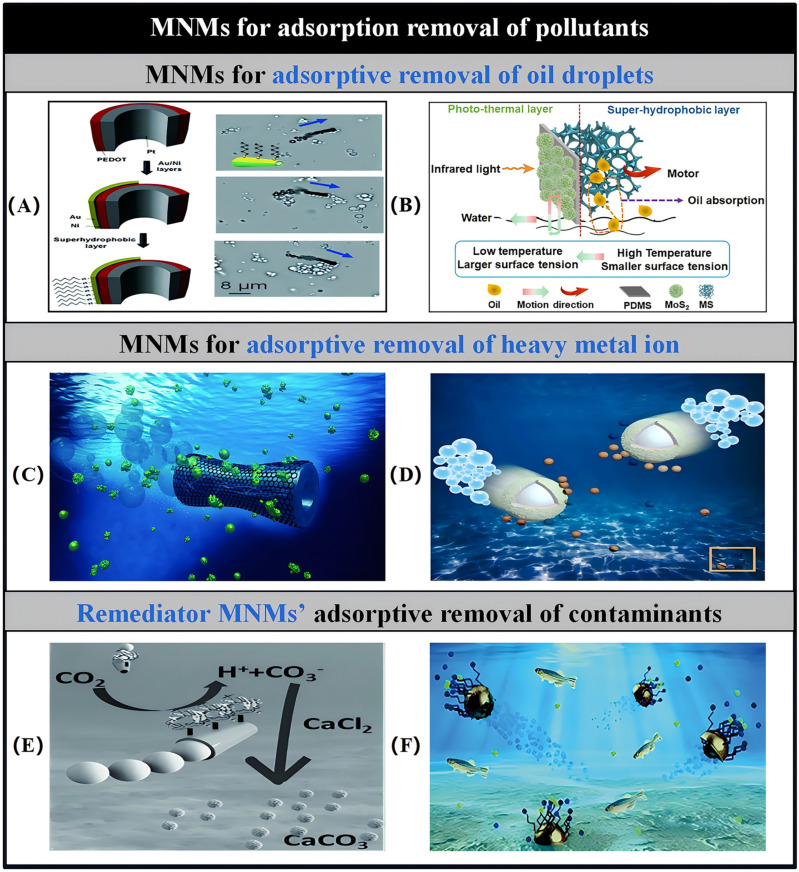


### 3.4. MNMs for Accelerated Pollutant Degradation

#### 3.4.1. Self-Propelled MNMs in Advanced Oxidation Processes for Pollutant Degradation

The phrase “Advanced Oxidation Processes” refers to a collection of chemical methods aimed at eliminating both organic and inorganic contaminants from water. These techniques utilize oxidative reactions involving hydroxyl radicals, ozone, H_2_O_2_, or UV light, specifically designed to target and decompose various harmful substances. In this context, H_2_O_2_ serves as a powerful oxidizing agent, presenting an innovative strategy for the removal of persistent pollutants. This methodology has been widely embraced in environmental applications for the breakdown of toxic compounds. The mobility of MNMs enhances the mixing and interaction between reactive agents and pollutants, leading to a more effective degradation process compared to stationary reactants. For example, Fe/Pt tubular MNMs, powered by H_2_O_2_ and produced through a convolutional technique, demonstrate the ability to degrade pollutants via the Fenton oxidation reaction, which holds promise in the realm of organic matter degradation. Under acidic conditions (pH 2.5) and with H_2_O_2_ as fuel, the outer Fe layer of the MNMs gradually converts to Fe^2^^+^, driving the Fenton reaction in conjunction with H_2_O_2_. The generated hydroxyl radicals then interact with rhodamine 6G, a model pollutant, leading to the disintegration of the organic ring structure into various oxidation products, which are eventually transformed into CO_2_. The dynamic motion of the MNMs results in pollutant removal rates approximately 12 times greater than those achieved with static MNMs [51]. Moreover, enhancing the thickness of the Fe layer can significantly improve degradation efficiency. Subsequent investigations by the same research team, utilizing 4-nitrophenol as a model pollutant, demonstrated that MNMs could remain in continuous motion for over 24 h and be stored for more than five weeks through multiple cleaning cycles. This finding indicates their potential for large-scale applications in the future [230]. Similarly, versatile zero-valent Fe/Pt Janus MNMs, powered by H_2_O_2_, have been utilized as mobile Fenton-like catalysts to degrade methylene blue, achieving complete degradation within 60 min of treatment [231]. Subsequently, researchers began exploring alternatives to Pt catalysts or employing adsorption-degradation configurations to enhance the overall efficiency of the Fenton MNM approach. For instance, moving hydrogels based on Prussian blue-reduced graphene oxide [232] and Pt-free cobalt ferrite MNMs [233] was proposed for the Fenton-mediated removal of methylene blue and antibiotics, respectively. Additionally, a zeolitic imidazolate metal–organic framework (ZIF)-Zn-Fe-Ag MNMs has been documented for the degradation of rhodamine B (RhB) via Fenton-like mechanisms (Figure 9A). The introduction of H_2_O_2_ led to improved adsorption of RhB on the ZIF-Zn-Fe MNMs matrix, attributed to the efficient motion and bubble formation of the MNMs. Simultaneously, the catalytic degradation of H_2_O_2_ at the Ag sites generates a considerable quantity of •OH radicals, which facilitate the oxidation of RhB both within the matrix and in the surrounding solution [234].

A trailblazing investigation led by Wang’s team demonstrated the effectiveness of bubble-propelled polymer/Pt composite MNMs in accelerating the oxidative detoxification of organophosphorus nerve agents (Figure 9B) [235]. In this system, H_2_O_2_ served as both the oxidizing agent and the fuel, while NaHCO_3_ and NaOH functioned as oxidation activators. The in situ production of peroxide anions, achieved without external stirring, enabled the conversion of nerve agents into p-nitrophenol. The key drivers of the oxidative detoxification process were the forced convection and enhanced mass transport of MNMs toward the reactive agents. Further studies revealed that cost-effective mobile MnO_2_ microspheres could also efficiently degrade methylene blue contaminants using H_2_O_2_ as the remediation agent, offering a similarly economical approach [236]. Currently, researchers are exploring NaBH_4_ as an alternative remediation agent and fuel source. For instance, a wastewater-based degradation method for 4-nitrophenol (4-NP) using MNMs has been developed (Figure 9C). In this process, Ti/Fe/Cr MNMs, decorated with Pd nanoparticles, were paired with the reducing agent NaBH_4_. When exposed to NaBH_4_, the Pd nanoparticles catalytically reduced 4-NP, driving the MNMs’ movement while converting the fuel into 4-aminophenol. The enhanced mobility of these MNMs allowed for pollutant removal up to ten times faster than their stationary counterparts [237]. Additionally, magnetic mesoporous CoNi@Pt nanorods displayed superior motility and facilitated the degradation of azo dyes in a NaBH_4_-rich solution [238].

To address the bioaccumulation and long-term persistence of organic pollutants, integrating multifunctional pollutant degradation processes that assist the oxidative functions of MNMs is essential. The MNMs-in-sponge system leverages the synergistic interaction between the sponge’s hydrophobic nature and the rapid pollutant degradation facilitated by embedded cobalt-ferrite (CFO) MNMs at its core [178]. This design results in efficient in situ degradation. The rapid degradation process stems from the bubble propulsion mechanism generated by the CFO, which enhances fluid mixing within the sponge and promotes outward fluid flow, leading to rapid fluid exchange. This contaminant removal platform operates effectively at low fuel concentrations (0.13%) and is both reusable and easily recyclable. Additionally, the MNMs-in-sponge system can efficiently degrade and treat large volumes of contaminants over multiple cycles within a short period. However, this approach is limited to addressing single-pollutant scenarios. Despite significant advances in synthesizing various MNMs for environmental remediation over the past decade, constructing pollutant-fueled MNMs capable of synergistically degrading multiple pollutants remains challenging. To address this limitation, Wang et al. developed laccase-powered Fe_3_O_4_@SiO_2_ MNMs [55]. With the assistance of lipase, these MNMs utilized pollutants as fuel to enhance the degradation of multiple contaminants. When activated by representative industrial pollutants such as phenols and PAHs, the Brownian motion of these MNMs was significantly enhanced, enabling efficient degradation of multiple pollutants within 40 min. This study expands the database of enzymatic MNMs and offers valuable insights into co-enzymatic catalysis, paving the way for broader applications in environmental remediation.

While the aforementioned pollutant treatment processes primarily focus on aquatic environments, the severity of soil contamination is increasingly evident. The limited migration efficiency of remediation agents in subsurface environments is a significant challenge in in situ soil remediation, and MNMs designed with bio-inspired properties offer a promising solution [239,240]. For instance, MNMs composed of natural pyrolusite and Fe_2_O_3_ have been employed as activators of peroxymonosulfate and H_2_O_2_ for the dynamic remediation of soil contaminated with polycyclic aromatic hydrocarbons. These bio-inspired MNMs exhibit significant microbubble generation, enabling rapid movement at ultra-low concentrations via bubble recoil, and serve as active heterogeneous catalysts to enhance the mass transfer process of remediation agents [240]. Moreover, visualization by coating the MNMs’ surface with the fluorescent material g-C_3_N_4_ demonstrated enhanced horizontal and vertical migration through the soil, driven by self-propulsion and bubble generation [239]. These bio-inspired MNMs present a novel strategy for enhancing the delivery of remediation agents in subsurface environments, offering substantial potential for efficient in situ soil remediation.
Figure 9(**A**) MNM-based advanced pollutant oxidation. (**a**) Tubular Fe/Pt rolled-up MNMs have been developed to eliminate organic pollutants via the Fenton reaction. These MNMs exhibit dual functionalities: the inner layer of Pt provides self-propulsion, while the outer Fe layer enables the in situ generation of Fe ions. The lower section illustrates the effect of Fe layer thickness on the removal efficiency of rhodamine. Reproduced from Ref. [51]. Copyright 2013, the American Chemical Society. (**b**) Self-propelling metal–organic framework/Ag Janus MNMs have been designed for the combined adsorption and Fenton degradation of rhodamine 6G. Reproduced from Ref. [234]. Copyright 2017, the Royal Society of Chemistry. (**B**) Accelerated oxidative detoxification of nerve agents utilizing MNMs, along with a schematic representation of the degradation process. Reproduced from Ref. [235]. Copyright 2013, the Wiley-VCH. (**C**) Activated Pd/Ti/Fe/Cr rolled-up MNMs were utilized for the degradation of 4-nitrophenol in wastewater, employing NaBH_4_ as both a reductant and a fuel source. Reproduced from Ref. [237]. Copyright 2016, the American Chemical Society. (**D**) TiO_2_/AuNPs/Mg MNMs, powered by water, were utilized for the photocatalytic degradation of both chemical and biochemical warfare agents. Reproduced from Ref. [241]. Copyright 2014, the American Chemical Society. (**E**) TiO_2_/Pt tubular MNMs, propelled both internally and externally, were utilized for the photocatalytic degradation of azo dyes. Reproduced from Ref. [242]. Copyright 2017, the American Chemical Society. (**F**) Plasmonic MNMs for the photocatalytic degradation of organic pollutants. Reproduced from Ref. [243]. Copyright 2016, the Royal Society of Chemistry. (**G**) Light-driven Au-WO_3_@C Janus MNMs for photodegradation of dyes. Reproduced from Ref. [31]. Copyright 2017, the American Chemical Society.
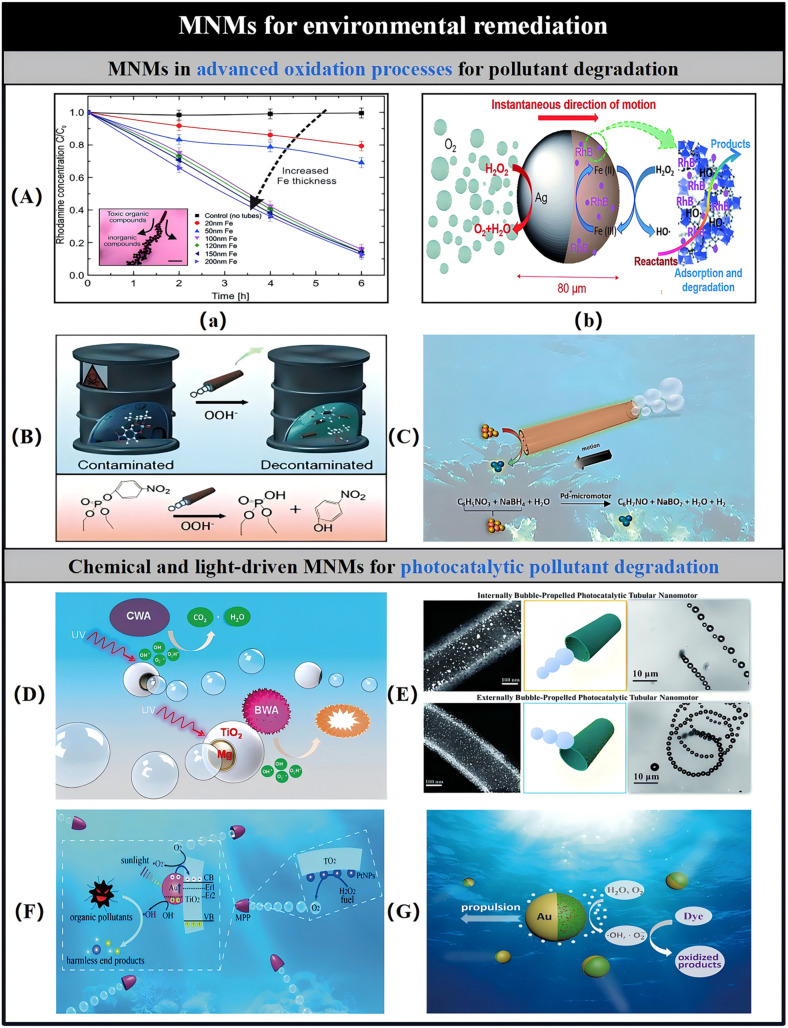


#### 3.4.2. Self-Propelled MNMs for Biocatalytic Pollutant Degradation

Enzyme-assisted pollutant removal offers a natural, non-invasive, and highly effective approach for eliminating contaminants. However, despite its promising potential, research on this method remains limited. A notable study demonstrated that biocatalytic MNMs created by filling pipette tips with SDS and a laccase solution effectively removed pollutants such as rhodamine black-T and 2-amino-4-chlorophenol. The movement of these MNMs was driven by the Marangoni effect, wherein laccase, functioning as the decontaminant, and SDS, acting as the propellant, were simultaneously introduced into the contaminated medium. This efficient distribution of laccase not only enhanced the fluid dynamics but also accelerated the biocatalytic detoxification process [244]. Similarly, MNMs enriched with enzymes derived from untreated Raphanus sativus radish tissues, containing both catalase and peroxidase, exhibited rapid movement within three minutes and demonstrated effective detoxification of pollutants such as 2-amino-4-chlorophenol, catechol, and guaiacol [245].

To further improve the biocatalytic degradation of pollutants, modifying MNMs through multiple enzyme combinations has been demonstrated as a feasible approach. For instance, Gao et al. synthesized novel MNMs that combined selectivity and catalytic activity by incorporating natural enzymes and nanoenzymes, modeled after natural halloysite nanotubes (HNTs) [246]. These MNMs feature a unique layered structure: an inner layer of MnO_2_ serves as a fuel-catalytic unit supplying continuous propulsion energy, while the outer layer of Lac/Fe-BTC@NiFe-LDH exhibits potent peroxide-based enzymatic activity and offers numerous active sites for laccase loading. The integration of natural enzymes with nanoenzymes enhanced the catalytic activity of the MNMs, enabling more sensitive detection and rapid degradation of minocycline (MC), a target water pollutant. At the same time, resource efficiency can be maximized by providing additional catalytic oxidation pathways to generate additional by-products while ensuring rapid and efficient pollutant removal. Therefore, Campos et al. developed a MnO_2_ tubular MNM modified using laccase, which could provide additional oxidative catalytic pathways for pollutant removal, thus enhancing the oxidation of organic pollutants [247]. The degradation efficiency of rhodamine B was enhanced by nearly 20% compared to unmodified MNMs. At the same time, these modified MNMs demonstrated efficient ammonia generation through the catalytic decomposition of urea in a short period of time, thus exemplifying potential application scenarios in the field of green energy generation. However, current MNMs designed around enzyme catalysis technology primarily feature complex 3D structures with limited surface accessibility for catalytic sites, necessitating higher fuel concentrations to maintain active motion. Consequently, enzyme-propelled 2D MNMs with increased catalytic sites have been developed. These 2D MNMs demonstrate efficient orthotropic behavior, capable of swimming against gravity with solution-provided buoyancy and exhibiting strong “on-the-fly” removal of methylene blue dye [120].

#### 3.4.3. Chemical and Light-Driven MNMs for Photocatalytic Pollutant Degradation

The use of light sources such as UV, VIS, and NIR offers a practical and biocompatible method for water purification and pollutant removal. Photocatalysis is considered an advanced water oxidation technique, where light and semiconductor metal oxides function as catalysts simultaneously. In some cases, an additional oxidizing agent, such as peroxide oxidizing, may be needed to promote the formation of free radicals, which can be used in subsequent pollutant degradation reactions [248]. Utilizing light to enhance or support the effectiveness of MNMs in environmental cleanup presents several advantages. Firstly, it eliminates the need for direct connection to external renewable energy sources. Secondly, it allows for precise control over both the movement of MNMs and the reaction speed. Finally, no toxic substances are required in the process [29,30]. The first generation of MNMs enabled photocatalytic degradation of a wide range of pollutants by combining UV light with the breakdown of peroxide fuel using a Mg/catalytic layer. This configuration employed TiO_2_, a highly efficient semiconductor, as the core material for MNM production. When exposed to UV light, TiO_2_ absorbs photons with an energy greater than its bandgap, causing electrons to shift from the valence band to the conduction band, thus generating positively charged holes. The resulting electron–hole pairs can either recombine, releasing heat, or interact with molecules like peroxides to form •OH radicals, which serve as potent oxidants [249]. An early approach was both biocompatible and environmentally friendly for degrading chemical warfare agents utilized Mg nanoparticles coated with TiO_2_/Au MNMs (Figure 9D). In this system, the TiO_2_ layer generates the oxidants necessary for pollutant mineralization under UV exposure. During the purification process, the Mg core not only drives the movement of the MNMs but also enhances fluid mixing, creating an efficient microsystem for photocatalytic purification. Near-complete degradation of nerve agent analogs, such as methyl paraoxon and bis(4-nitrophenyl) phosphate, by MNMs demonstrates their potential for use in neutralizing pathogenic bacteria [241].

A different method for water remediation involved subjecting coaxial TiO_2_-Pt/Pd-Ni microtubes to UV, visible, and natural light. Under both visible and natural light, the heterostructure achieved 100% degradation efficiency of rhodamine B. This multi-component system has the potential to harness external energy sources such as magnetic fields, ultrasound, or H_2_O_2_ fuel for enhanced propulsion and performance [250]. Alternative configurations rely on the use of peroxide as a co-reagent, in conjunction with light irradiation and MNM movement. For example, Pt nanoparticles were placed on the internal and external surfaces of TiO_2_ tubular MNMs (Figure 9E). The superhydrophilic properties of TiO_2_ enabled the MNMs to move in a low-concentration H_2_O_2_ solution of just 0.05%, without requiring surfactants. The system demonstrated fluid mixing and photocatalytic abilities, effectively degrading rhodamine B under UV light [242]. Similarly, both the inner and outer surfaces of TiO_2_ microshells were coated with AuNPs and PtNPs, through template-assisted and aqueous-phase synthesis, respectively (Figure 9F). Under anaerobic conditions, the MNMs utilized artificial sunlight to degrade H_2_O_2_ fuel, producing O_2_ bubbles that propelled the system and efficiently degraded methylene blue, rhodamine B, and methyl orange. The AuNPs on the TiO_2_ shells improved electron–hole separation efficiency and adsorption, boosting photocatalytic performance [243]. CdS quantum dots were later introduced as a substitute optical catalyst for TiO_2_. The peroxide-driven CdS/polyaniline Pt microtubule demonstrated high efficiency in degrading bisphenol A under solar radiation. Here, reactive oxygen species such as •OH and superoxide ions were key intermediates in the degradation process. Photons with energy exceeding the CdS bandgap excited valence-band electrons to the conduction band, creating positively charged holes. These electrons could then react with water or •OH to produce reactive radicals [251].

To promote more eco-friendly environmental remediation, light has been utilized not only to propel MNMs but also to break down pollutants. Among the many water quality pollutants, endocrine-disrupting chemicals are considered the most dangerous compounds to marine organisms and human health. MNMs constructed with MnO_2_/TiO_2_-metal materials have strong self-propulsion properties and photoactivity for on-the-fly photocatalytic degradation and removal of these chemical threats [252,253]. TiO_2_ microtubule arrays, created by electrochemical anodization of Ti sheets, are driven by diffusiophoresis when exposed to UV light [253]. Water-powered TiO_2_/Pt Janus MNMs exhibit either aggregated or dispersed behaviors, and their movement can be wirelessly controlled by adjusting the UV light’s “on–off” switch, intensity, and pulse or continuous irradiation modes. The efficient propulsion of these particles in water results from light-induced self-electrophoresis, driven by asymmetric water oxidation and reduction reactions on their surfaces, and is used for the degradation of rhodamine B [254]. Similarly, water-driven TiO_2_/Au Janus MNMs have been observed to degrade dye pollutants under UV light. Notably, the speed of MNMs movement significantly increased when dyes like methyl blue, cresol red, and methyl orange were present in the solution. This enhanced motion is attributed to self-electrophoresis caused by the photocatalytic degradation of the dyes on the MNMs’ asymmetric surface. Under UV radiation, electrons in the valence band of TiO_2_ are excited to the conduction band, allowing for efficient electron transfer from TiO_2_-Au. This process not only reduces electron–hole recombination but also extends the operational lifespan of MNMs. The separated electron–hole pairs lead to the creation of reactive superoxide radicals, which subsequently interact with dye molecules, causing their oxidative breakdown into mineralized products [98,255]. Additionally, light-activated Au-WO_3_@C Janus MNMs have been employed for the swift photocatalytic degradation of sodium-2,6-dichloroindophenol and rhodamine B (Figure 9G). In this case, diffusiophoresis serves as the primary driving force for the MNMs’ movement. Photodegradation products were observed to accumulate more on the WO_3_ side than the Au side. The porous nanoscale structure of WO_3_@C microspheres facilitates the diffusion of reaction products, inducing osmotic flow from areas of low solute concentration to areas of higher concentration on the MNMs’ surface, thereby propelling them forward. Simultaneously, the mineralization of the dye generates oxidation radicals [19]. Moreover, star-shaped AgCl MNMs have shown the capability to degrade various organic dyes and suppress bacterial growth. Upon exposure to UV light, Ag^+^ in AgCl is reduced to metallic Ag, which creates a local electrolyte gradient that propels the MNMs through self-diffusiophoretic motion [256]. Furthermore, in addition to the means by which externally applied fuels and externally applied fields can enable environmental remediation of MNMs, naturally occurring contaminant molecules in the environment can also be used as chemical fuels in the process. For example, Mushtaq et al. demonstrated the catalytic degradation of organic compounds using the magnetoelectric properties of core–shell nanoparticles made from cobalt–bismuth ferrite (CFO-CBO). By integrating the magnetostrictive properties of CFO with the multiferroic nature of BFO, they created MNMs that respond to magnetoelectric stimulation. When subjected to an external magnetic field, these MNMs facilitate the breakdown of organic pollutants through an advanced oxidation process, effectively purifying water. Notably, the process does not require the addition of any sacrificial molecules or external catalysts [257]. At the same time, enzymes can be used to advance MNMs for environmental remediation. For example, Orozco et al. introduced a novel micromotor-based water quality testing strategy. This strategy is based on changes in the driving behavior of the artificial biocatalytic microswimmer in the presence of aquatic pollutants [193].

### 3.5. Self-Propelled MNMs for Pathogenic Bacteria Removal

To achieve effective antimicrobial therapy, researchers have begun to develop strategies based on MNMs. For example, chitosan/Mg Janus MNMs were created to eliminate Escherichia coli bacteria present in contaminated seawater or freshwater (Figure 10A). The chitosan coating on these Mg-based MNMs facilitates their movement in water, enabling them to make contact with and destroy bacterial cells through the action of the outer chitosan layer. The death of the bacteria is promoted by mechanical interactions between the chitosan on the MNMs’ microstructure and the bacterial cells [258]. Additionally, an innovative and environmentally friendly approach involves the use of lysozyme-modified, ultrasound-propelled nanowire MNMs. Lysozyme possesses natural antimicrobial properties that can break the glycosidic bonds in the peptidoglycan layer of bacterial cell walls. Coupled with the rapid movement of the MNMs, this significantly enhances their ability to kill bacteria. As a result, these MNMs have achieved the rapid and effective elimination of 69–84% of Gram-positive Micrococcus lysodeikticus bacteria within five minutes, indicating their considerable potential for future nanotechnology-based water purification methods (Figure 10B) [34]. Similarly, TiO_2_/Mg MNMs have proven effective in inactivating Bacillus globigii spores [250]. Furthermore, the antimicrobial properties of Ag can be utilized to kill bacteria by decorating Mg/Fe MNMs with Ag nanoparticles [259]. The movement of these MNMs aids in transporting the released Ag^+^, while the internal ferromagnetic layer allows for convenient remote guidance and removal. Likewise, zeolite/Ag MNMs have also demonstrated effectiveness in eliminating motile Escherichia coli bacteria (Figure 10C) [221].

## 4. Limitations of MNMs in Environmental Remediation

MNMs exhibit significant potential due to their active mobility and programmable coordination in contaminant detection and degradation, demonstrating broad efficacy in removing a range of contaminants with varying toxicity levels (Figure 11). However, practical applications present considerable challenges. (i) The primary challenge is scalability [260,261]. Although MNMs have shown impressive results in laboratory-scale experiments, scaling up production for real-world environmental remediation remains difficult. While some MNMs can be synthesized using low-cost biological or mineral materials, most require expensive components, complex equipment, and multi-step processes for production. Therefore, developing manufacturing methods that enable industrial-scale production while ensuring consistent quality and performance is essential. (ii) Additionally, the movement of MNMs in dry, solid environments is significantly hindered by friction. On the microscale, van der Waals forces cause interfacial adsorption and friction that far outweigh gravitational and inertial forces, impeding autonomous propulsion. Research on improving MNM propulsion in solid media remains limited, underscoring the need for more robust MNMs to enhance mobility and pollutant removal efficiency in such conditions. (iii) Swarm control technology for MNMs also requires substantial development, including intra-swarm communication, autonomous adaptation to complex environments, cooperative division of labor, and self-regulation within the swarm. (iv) Energy supply is another significant limitation, as MNMs’ movement often relies on external energy sources such as light, chemical fuels, or magnetic fields. In turbid water or soil environments, light or magnetic field transmission may be obstructed, and controlling the concentration and distribution of chemical fuels precisely is challenging, with the risk of causing secondary pollution. Exploring self-sustaining systems and efficient catalytic materials is crucial to enhance MNMs’ energy independence in various conditions. (v) The efficiency of movement and transport is a notable limitation in aquatic environments. MNM propulsion, driven by diffusion, light, chemical reactions, or magnetic fields, can be affected by water viscosity, fluidity, and pollutant concentrations. Turbid water with suspended particles may obstruct or slow MNMs’ movement, decreasing remediation efficiency. Therefore, further optimization of MNM structures is necessary to maintain stable and effective movement in complex media. (vi) MNMs must also address challenges related to selectivity and resistance to interference in multi-pollutant environments. Although MNMs are designed to target specific pollutants, real-world conditions often involve complex pollutant mixtures that may interact physically or chemically, interfering with MNMs’ recognition and catalytic processes and reducing remediation effectiveness. Enhancing MNMs’ adaptability through multifunctional composite designs and surface functionalization is essential. (vii) In complex natural settings, stability and persistence are also significant issues. Variations in environmental conditions such as pH, temperature, and salinity can degrade MNMs’ structures and functions. Additionally, collisions with other particles can cause structural damage or surface property changes, impacting stability and lifespan. These challenges must be addressed by optimizing material design and improving resistance to environmental stresses. (viii) Ensuring the safe recycling and degradation of MNMs after their mission is a potential concern. Without proper degradation mechanisms, MNMs remaining in the environment could contribute to secondary pollution. Therefore, integrating degradability and recyclability into MNMs’ designs is necessary to prevent new environmental impacts post-remediation.

## 5. Conclusions and Perspectives

The innovative application of synthetic MNMs holds significant promise for advancing solutions in environmental remediation. The use of self-propelled MNMs marks a transformative shift in this field, significantly improving the effectiveness of traditional methods through the active movement of materials. This review explores both the potential and challenges associated with the practical application of these recent innovations. It highlights how mobile reactors or receptors can significantly enhance the effectiveness and speed of environmental remediation and monitoring processes. The innovative features of synthetic MNMs that drive these promising environmental applications are examined. Additionally, the article discusses the current opportunities and challenges associated with utilizing these functional MNMs to improve various environmental solutions. Since their inception, MNMs have shown considerable promise in water toxicity assessment and the acceleration of environmental cleanup processes. First, we illustrate how MNMs can adapt their movement in response to various contaminants, a crucial factor for the efficient execution of remediation tasks. From initial motion-based detection techniques to contemporary setups incorporating fluorescent nanoparticles and dyes, MNMs offer substantial capabilities for identifying threats, monitoring remediation efforts, and functioning in hard-to-reach environments. Second, this paper emphasizes the role of MNMs as mobile reactors for water purification, particularly focusing on adsorption processes and enhanced degradation of contaminants. This area of research has gained momentum following the groundbreaking study on MNM-assisted Fenton oxidation for oil removal. Recent advancements in pollutant removal through adsorption have concentrated on exploring nanomaterials with large surface areas, such as graphene, or combining these materials with effective catalysts for greenhouse gas capture. Currently, research in advanced pollutant treatment using MNMs is progressing rapidly, addressing crucial issues like scalability and reusability, which are vital for the practical application of these methods. However, many of these strategies still rely on MNMs (Mg/Zn) that are propelled by environmentally harmful H_2_O_2_ fuel with a short operational lifespan. Consequently, researchers are investigating new, eco-friendly propulsion mechanisms based on light or ultrasound. Notably, substantial progress has been made in light-driven MNMs, where UV light and sunlight play dual roles. They not only provide energy for MNM propulsion but also generate free radicals that aid in pollutant degradation. In summary, this review highlights the extraordinary potential of dynamic MNMs as effective agents in combating bacterial threats [262].

Where are we headed? The exploration of MNMs as a solution to environmental challenges represents an emerging field of research. Despite significant progress achieved to date, it is clear that further collaborative efforts are essential to convert these innovative proof-of-concept studies into effective, large-scale environmental applications (Table 3).

(i) Scalability: The synthesis process of some functionalized MNMs involves complex, multi-step procedures that hinder scalability and commercial production. Future research should prioritize simplifying the synthesis of MNMs, potentially utilizing 3D printing, and integrating automated manufacturing with modular designs for scalable and cost-effective production. (ii) Frictional Hindrance: The properties of friction in solid environments can significantly impede the kinematic behavior of MNMs, thereby diminishing their repair efficiency. To mitigate interfacial adsorption and friction effects, low-friction materials can be applied to coat and lubricate the surfaces of MNMs, thereby improving their kinematic performance in solid environments. Additionally, developing adaptive propulsion systems with integrated sensors and feedback control will enable MNMs to autonomously adjust their propulsion mode according to environmental conditions, thus enhancing mobility. (iii) Swarm Control: The load capacity of individual MNMs is limited due to their small size, making the development of swarm control techniques essential. The principles of velocity matching, cohesion, and coherence can be employed to facilitate coordinated movement among individual MNMs in a swarm, ensuring stable movement patterns and improving their capacity for complex tasks. (iv) Sustained Energy Supply: The conversion rate of external energy sources can be reduced by environmental constraints in real-world contaminated settings. Developing pollutant-fueled power systems or hybrid energy systems that enable MNMs to switch between energy sources under varying conditions could overcome these limitations, ensuring continuous operation. (v) Motion and Migration Efficiency: The motion efficiency of MNMs can be affected by factors such as water viscosity and pollutant concentration. Biomimetic design improvements, such as streamlined or helical structures, can reduce drag and enhance propulsion in aquatic environments, while concave and convex surface features provide better grip and mobility in soil. The negative impact of salt on the kinematic properties of MNMs that do not utilize bubble propulsion can be attributed to several factors, including an increase in ionic strength, an elevation in viscosity, a reduction in the rate of chemical reactions, and alterations in surface properties. To address these issues, the motility and adaptability of MNMs in high-salt environments can be enhanced by optimizing the solution environment (lowering the salt concentration), surface modification (introducing salt-resistant coatings), improving the actuation mechanism (employing magnetic or optical actuation), increasing the catalytic efficiency (using salt-resistant catalysts), and developing salt-responsive materials (salt-sensitive polymers). (vi) Selectivity and Interference Resistance: The co-existence of multiple pollutants can result in interactions that interfere with the selective recognition and catalytic degradation process of MNMs. By incorporating specific functional groups or molecular recognition elements on MNMs’ surfaces, target pollutant detection can be enhanced, while non-target adsorption and interference can be minimized. Furthermore, integrating diverse functional catalysts and adsorbent materials within MNMs enables simultaneous treatment of multiple pollutants, reducing efficiency losses due to competition. (vii) Stability and Durability: Dynamic pollutant environments can lead to structural and functional degradation of MNMs, reducing their stability and lifespan. Integrating corrosion- and oxidation-resistant composites into MNMs can improve their stability under varying pH, temperature, and salinity conditions. Additionally, protective coatings or self-repairing materials can be added to MNMs’ surfaces to minimize mechanical damage and chemical erosion, thus extending their service life. (viii) Degradability and Safety: MNMs may pose potential environmental risks after completing their remediation tasks. To mitigate this, biodegradable or environmentally friendly materials should be used in MNM fabrication, allowing them to degrade naturally after their task is complete. Additionally, in-built self-destruction or disintegration mechanisms can be designed to trigger decomposition under specific conditions, such as exposure to particular light, chemicals, or temperature, facilitating rapid breakdown and reducing environmental residues. Moreover, specialized recycling systems should be developed to collect MNMs post-use through magnetic or electrical guidance.

In the near future, it is expected that multifunctional, self-regulating MNMs will be developed, enabling them to perform complex tasks such as detecting, isolating, and neutralizing toxic pollutants and chemical threats; identifying sources of hazardous substances; or delivering sensors to distant and challenging environments. Groups of MNMs can be swiftly assembled for urgent scenarios, allowing for the mapping of toxic pollutant dispersion across extensive areas or accelerating environmental remediation initiatives. We hope this review will foster comprehensive research into the environmental applications of MNMs.

## Figures and Tables

**Figure 1 micromachines-15-01443-f001:**
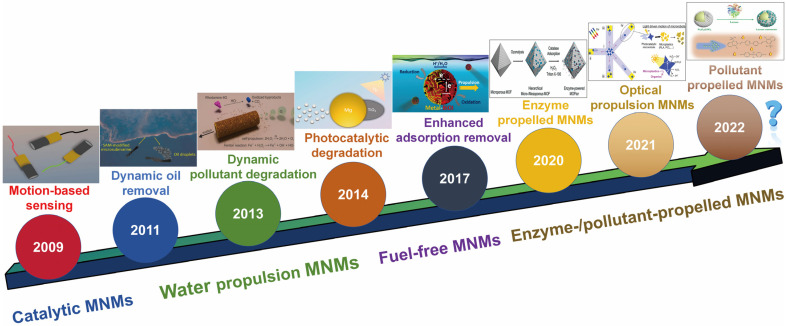
Timeline showing the progress in the use of self-propelled MNMs for environmental applications. Top part in 2011: reproduced from Ref. [50]. Copyright 2012, the American Chemical Society. Top part in 2013: reproduced from Ref. [51]. Copyright 2013, the American Chemical Society. Top part in 2017: reproduced from Ref. [29]. Copyright 2017, the American Chemical Society. Top part in 2020: reproduced from Ref. [54]. Copyright 2020, the American Chemical Society. Top part in 2021: reproduced from Ref. [53]. Copyright 2021, the American Chemical Society. Top part in 2022: reproduced from Ref. [55]. Copyright 2022, the Elsevier Ltd. All rights reserved.

**Figure 2 micromachines-15-01443-f002:**
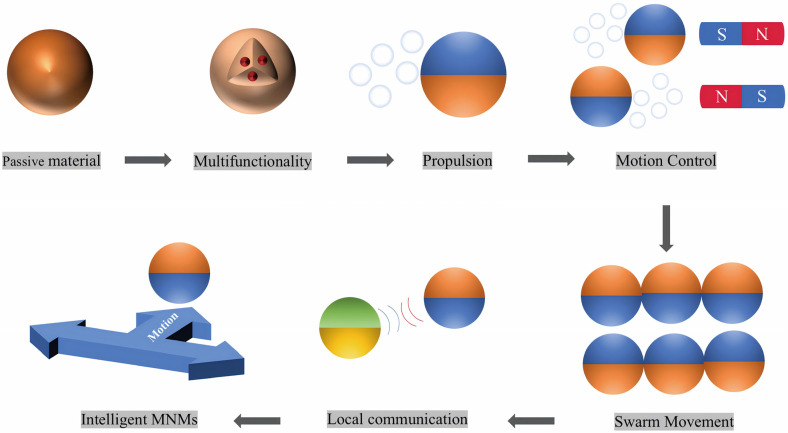
From passive materials to intelligent MNMs. After designing MNMs from micromaterials, several key steps are required to transform them into intelligent MNMs with autonomous movement capabilities. First, they must be endowed with multifunctionality, enabling them to carry out multiple specialized tasks in complex fluid environments. Second, propulsion mechanisms must be incorporated, allowing them to move autonomously by consuming chemical fuels or responding to applied fields. Third, motor control, such as chemotaxis, must be integrated, enabling MNMs to adaptively respond to external stimuli like chemical gradients, light (phototaxis), or magnetic fields (magnetotaxis). Fourth, swarm behavior should be developed to allow MNMs to move synergistically, enhancing processing efficiency or performing tasks beyond the ability of individual MNMs. Finally, local communication systems must be established, allowing MNMs to coordinate and exchange environmental information with neighboring units in a synchronized manner.

**Figure 3 micromachines-15-01443-f003:**
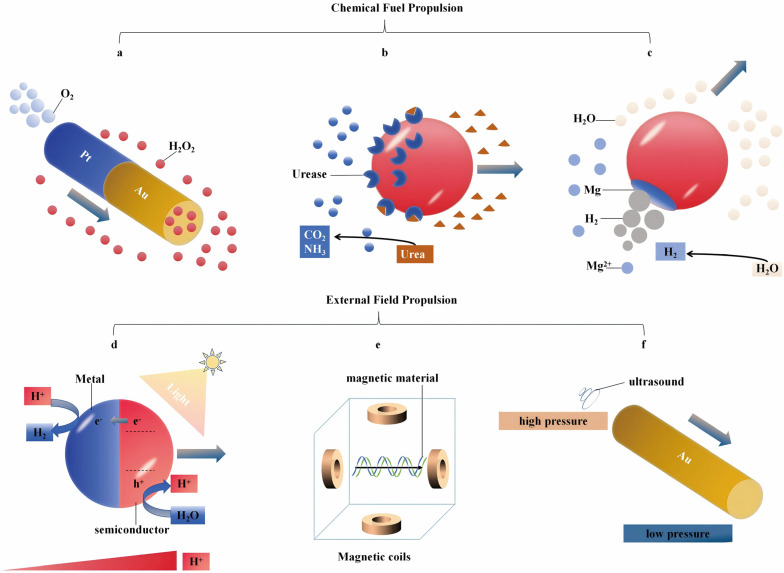
Motion mechanisms of fuel-driven and externally driven MNMs. (**a**) Pt/Au MNMs facilitate the breakdown of H_2_O_2_ fuel, leading to the generation of O_2_ bubbles that propel them forward with significant thrust upon bubble release. (**b**) Enzyme-driven MNMs catalyze the hydrolysis of substrates such as urea through uneven enzyme (urease) immobilization, creating a concentration gradient of the product that drives their movement via a self-phoresis mechanism. (**c**) MNMs made from reactive metals like Mg react with water during propulsion, producing a flow of H_2_ bubbles while simultaneously degrading the Mg core, thus advancing the MNMs. (**d**) Janus MNMs, composed of metal and semiconductor materials, are activated to move by the formation of electron–hole pairs (e^−^-h^+^) within the semiconductor under illumination. In this scenario, oxidation and reduction reactions involving H_2_O occur on opposing sides of the MNMs, resulting in a proton gradient and a locally generated electric field, which facilitate self-electrophoresis. (**e**) Magnetic helical MNMs can transform the rotational motion produced by a pair of orthogonally oriented electromagnetic coils into precise and controllable translational movement when subjected to a rotating magnetic field. (**f**) MNMs migrate from areas of high pressure to those of low pressure in response to ultrasonic waves.

**Figure 4 micromachines-15-01443-f004:**
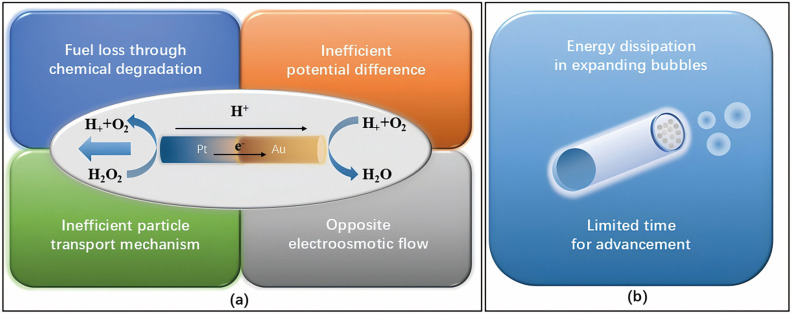
Reasons for the inefficiency of MNMs during self-electrophoresis mechanism and bubble recoil mechanism. (**a**) In the self-electrophoresis propulsion mechanism, the four energy loss stages of MNMs during the conversion process. (**b**) In the bubble recoil propulsion mechanism, MNMs have two stages of energy loss during the conversion process.

**Figure 5 micromachines-15-01443-f005:**
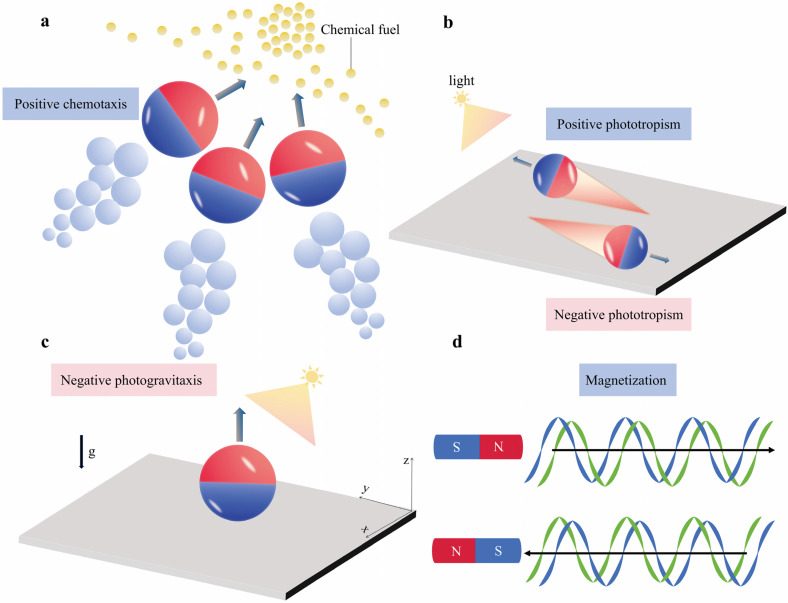
Convergent motion of MNMs. (**a**). Bubble-driven Janus MNMs migrate toward areas with elevated concentrations of chemical fuel. (**b**). Light-responsive MNMs adjust their motion, either approaching or moving away from the light source. (**c**). In the presence of light, MNMs exhibit negative photogravitaxis by moving in a direction counter to gravity. (**d**). Magnetic helical MNMs display altered kinematic behavior when subjected to a reversal of magnetic polarity.

**Figure 6 micromachines-15-01443-f006:**
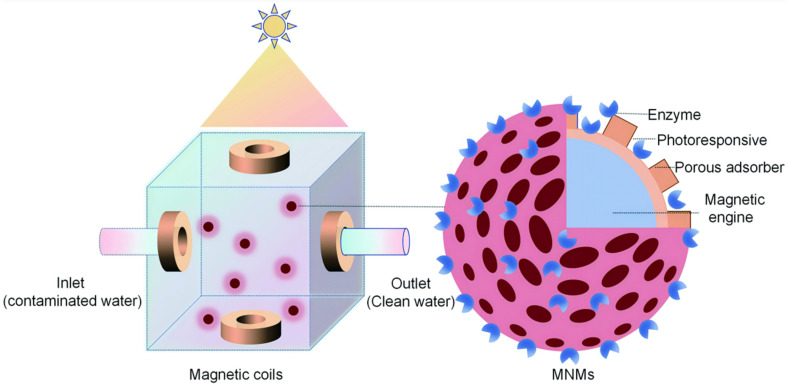
MNM-based environmental remediation system.

**Figure 10 micromachines-15-01443-f010:**
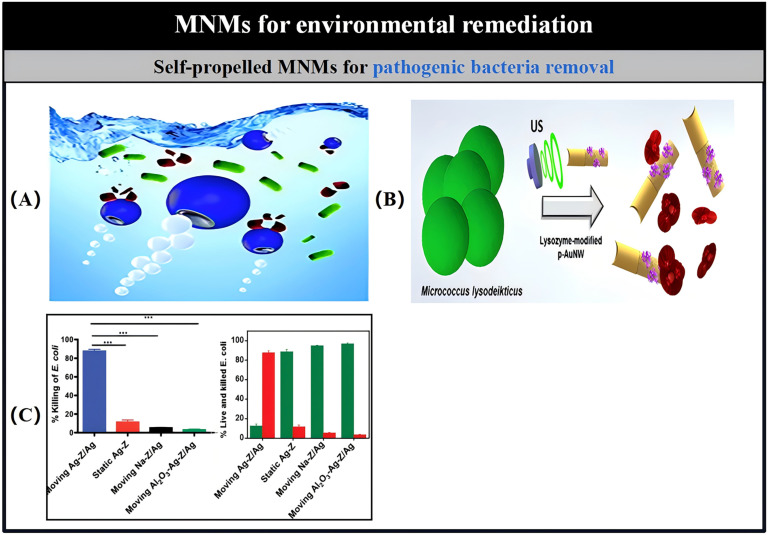
Self-Propelled MNMs for Pathogenic Bacteria Removal. (**A**) Water-powered Mg/Au/PLGA/Alg/Chi MNMs for Escherichia coli bacteria killing. Reproduced from Ref. [258]. Copyright 2017, the Royal Society of Chemistry. (**B**) Ultrasound propelled lysozyme MNMs for killing bacteria. Reproduced from Ref. [34]. Copyright 2015, the American Chemical Society. (**C**) Zeolite/Ag Janus MNMs for biological warfare agent’s detoxification. Reproduced from Ref. [221]. Copyright 2015, the Wiley-VCH.

**Figure 11 micromachines-15-01443-f011:**
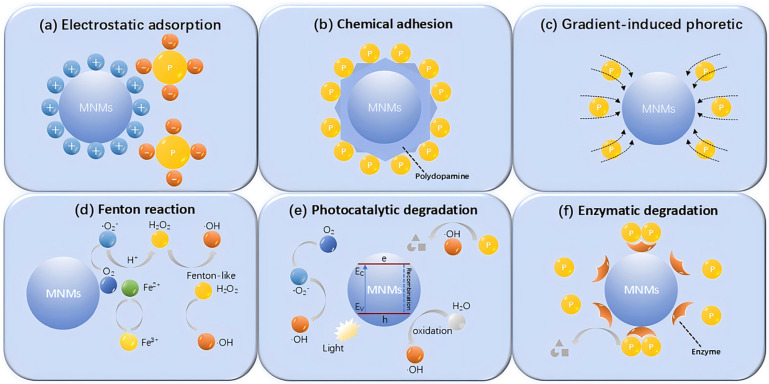
Mechanisms of pollutant removal and degradation by MNMs.

**Table 1 micromachines-15-01443-t001:** Preparation of MNMs.

Fabrication	Technique	Structure	Advantages	Limitations
Electrochemical	Membrane template-assisted electrochemical deposition	Nanowires, Nanorods, Microtubes	Low cost, prepared from inorganic or organic materials	Large-scale production and MNMs’ geometry are limited by the characteristics of the membrane (pore size, shape)
Chemical	Polymerization, hydrothermal reaction, precipitation, solvent extraction and evaporation, functionalization, reactive ion etching	Symmetric and asymmetric MNMs	Simple equipment, low-cost and large-scale production	Reproducibility
Physical	Physical vapor deposition (sputtering, evaporation, atomic layer deposition), lithography	Anisotropic structures, coatings for Janus structures	Highly controllable and reusable	Expensive equipment
Biohybrid	Bio-templated MNM fabrication, bio-hybrids (microorganisms modified with MNMs)	Organic and inorganic microstructures, self-motile structures for hybrid MNMs	Low cost, sustainability, and abundance of bio-templates	Short MNM lifespan
Self-assembly	Layer-by-layer assembly, macromolecular assembly, shape transformation	Vesicles, Janus capsules, polymersomes	Simple equipment, low-cost, sustainability, versatility, bio- inspired and biodegradable materials preparation	Mostly fuel driven
3D printing	Fused deposition modeling, selective laser sintering, direct ink writing	3D and 4D structures	Low-cost and large-scale production, high control and reproducibility	Limited resolution and range of suitable materials

**Table 2 micromachines-15-01443-t002:** Mechanisms for propulsion of MNMs.

Propulsion Approach		Performance	Span	Safety	Limitations
External Field	Magnetic fields	Enables precise 3D movement	Better persistence; MNMs can sustain motion guided by the external field	Good biocompatibility; strong magnetic fields may affect the human body	Electromagnetic drive with low energy efficiency and limited working space
	Light	On–off control and directional motion can be realized with fast motion speeds		UV light is harmful to human body; other light is basically safe	Driving in liquids requires high power; limited light focus size and range of motion
	Ultrasound	Fast motion response		Safer	Lack of operational precision
	Electric field	Chemotaxis		Strong electric fields may have an effect on the human body	Small range of motion
Chemical	Bubble	Dependent on bubble size or vibration frequency		Related to the way the bubbles are generated	Lack of operational precision and efficiency
	Self-phoresis	Fast movement speed	Poor sustained performance; as chemical fuels are gradually consumed, motility performance will decline	Enzyme-initiated biocatalytic reactions are safe and biocompatible, but H_2_O_2_ is harmful	Dependent on fuel concentration; poor motion continuity; uncontrollable motion direction and accuracy
	Ionic diffusiophoresis	Dependent on catalytic reaction rate, the strength of the ion concentration gradient, and the electrodynamic effect on the particle surface	Affected by fuel consumption, catalyst stability, and environmental conditions	Depends on the biocompatibility of the fuel, catalyst, and its by-products	In solutions with high salt concentrations, the ion shielding effect weakens the concentration gradient and affects the movement efficiency [115,116,117]
	Enzymatic reactions	The speed is relatively stable and can maintain the movement for a longer period of time	Depends on the catalytic efficiency of the enzyme, rate of diffusion of reaction products	Better biocompatibility	Enzymes are easily inactivated by environmental changes (temperature, pH), resulting in cessation of movement [111,118,119]
Biologically driven		Relying on the motor properties of individual organisms	Influenced by biological cell activity	Good biocompatibility	Cells, bacteria, etc. need specific nutrient solutions as well as an environment to survive

**Table 3 micromachines-15-01443-t003:** Challenges and strategies for MNMs.

Challenges	Strategies
Scalability	Laser-based 3D printing
Frictional hindrance	Surface modification and lubrication strategies; adaptive advancement mechanisms
Swarm control	Applying speed matching, cohesion, and consistency rules
Sustained energy supply	Pollutant-fueled power systems or hybrid energy systems
Motion and migration efficiency	Optimized structural design
Selectivity and interference resistance	Multi-functional composite design; surface modification and functionalization
Stability and durability	Protective coatings or self-healing materials
Degradability and safety	Degradable material design; self-destruction or disintegration mechanisms; recyclable systems

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
