# Peer review of "Application of Micro/Nanomotors in Environmental Remediation: A Review"

_micromachines, 2024, doi:10.3390/mi15121443_

Round 1

Reviewer 1 Report (New Reviewer)

Comments and Suggestions for Authors

Referee Report “Application of Micro/Nanomotors in Environmental Remediation: A Review” by Tao He, Shishuo Liu, Yonghui Yang and Xue-Bo Chen

This review paper provides a thorough examination of self-propelled and externally propelled micro/nanomotors (MNMs) and their innovative applications in environmental remediation. It effectively highlights the transformative potential of MNMs in enhancing the efficiency of conventional remediation techniques through their unique capability for active motion. Key areas of focus include the detection of environmental indicators, pollutant removal, and bacterial elimination. The review emphasizes how the dynamic motion of MNMs facilitates both detection and remediation processes and explores the multifunctionality of these systems, which can be actuated not only by chemical fuels but also through stimuli such as light, magnetic fields, electricity, and ultrasound.

The paper also delves into the challenges and limitations associated with MNMs, including issues related to scalability, mobility in solid media, swarm control, energy supply in complex environments, pollutant selectivity, material stability, degradability, safety, and eco-friendliness. The authors propose strategies to overcome these barriers, offering valuable insights into the current state of MNM technology.

This review stands out for its thorough analysis and breadth of coverage. The authors have meticulously collected and synthesized studies across various domains of environmental applications, offering a well-rounded perspective on the potential of MNMs. The inclusion of challenges and actionable suggestions for future research is particularly very valuable, providing a roadmap for advancing the practical utility of MNMs in real-world scenarios.

While the overall quality of the work is very good, there are few minor points that could benefit from further refinement to enhance the clarity and depth of the content. I would recommend publication, after revising the following issues:

-        Some MNM  preparation techniques are missing such as the ones based on material patterning through lithographies (e.g. photolithography) or reactive ion etching techniques which are quite versatile.

-        Line 151. I suggest revising the sentence for greater accuracy: “In contrast, urease and glucose oxidase facilitate the hydrolysis of more biocompatible substrates, specifically urea and glucose, respectively.” Urease indeed promotes the hydrolysis of urea, but glucose oxidase catalyzes the oxidation of glucose and not its hydrolysis. Alternatively, you could generalize the statement by saying that urease and glucose oxidase facilitate the catalysis of more biocompatible substrates.

-        Line 156, connect better the sentence with the flow of the paragraph to increase clarity and readability, just saying that in some contexts the catalyzed reaction consumes the MNMs and the fuel as in the case of Mg-H2O2.  

-        Line 160, just consider that some fuels can be also salt cations, such as in the case of chemically propelled motors based on ion-exchange

-        Correct caption “Table 2. Motion control of MNMs of MNMs.”

-        In Table 2, it is crucial to include one of the primary limitations of (photo)chemically propelled MNMs, specifically for those propelled by self-electrophoresis and ionic diffusiophoresis (even those activated by enzymes): their lack of tolerance to salts. The self-generated electric fields responsible for their motion are significantly screened by salt ions, which drastically reduces their motion velocity. In recent years, some nanomotors based on ion exchange mechanisms have been explored to overcome this issue. These nanomotors demonstrate greater salt tolerance, even though they still produce electric fields. However, their motion remains effective because the generated electric field is proportional to the gradient of salts in the medium. Relevant articles discussing this limitation, including those on immobilized motors acting as pumps and the collective behavior of ion-exchange motors can be cited (Journal of Chemical Physics 144, 124702 (2016); Nat Commun 13, 2812 (2022); Faraday Discuss., 2024,249, 424-439).

-        Table 2. Better wording for the following sentence:” Poor sustained performance, as chemical fuels are gradually consumed, athletic performance will suffer”. Alternative: Poor sustained performance, as chemical fuels are gradually consumed, motility performance will decline.

-        Consider introducing the effect of temperature on speed control in addition to fuel concentration, as it can significantly enhance motility. Notable studies on temperature effects on catalysis from Josep Wang’s group can be cited, along with work on photothermal effects from light-activated plasmonic materials (e.g., Promising Advances in Physically Propelled Micro/Nanoscale Robots in Nano Material Sciences or a recent study in ACS Appl. Nano Mater. 2024, 7, 23321−23336).

-        Connect better the paragraph starting in line 187 in the context of speed control.

-        Improve transitions between natural examples and MNM applications (line 339-340). Consider adding a bridging sentence.

-        Consider adding a reference on swarming or collective behavior of ion-exchange motors (e.g. Faraday Discuss., 2024,249, 424-439 or the studies from Palberg group).

-        Line 467, I would say the underpotential deposition of Ag on Au/Pt segments, since the Ag is deposited more at the Au side and increasing the velocity of the bimetallic rod.

-        Line 517 should say ricin instead of cricin

-        Line 530 has a misspelling of plectin, it should say lectin

-        Line 693, elaborate better connection since another example is described.

-        Line 717-719, the sentence “The capacity of Hg2+ to adsorb onto these mobile microtubules is attributed to its high specificity and strong affinity for the mismatched T-T pairs in DNA sequences” is disconnected in the flux of the context.

-        Line 765. “12 times greater than those achieved with static MNMs50”. The reference is not 50, it is 51.

-        Line 795. The fuel should be NaBH₄ instead of NaOH.

-        Line 845 has a misspelling, it should say lacasse instead of acase.

-        Elaborate better the connection between the lines 863-865 and the corresponding studies (refs 210, 211). The current phrasing may cause confusion, as it appears to continue describing the studies from reference 210.

-        Line 878, better use the wording “photocatalysis” instead of photolysis.

-        Line 906, add a reference regarding the use of perovskites in the MNMs.

-        Lines, 906, 920. Indicate the radical OH with its corresponding dot to denote its radical character.

-        Lines 1094-1095. Elaborate better the sentence since it seems that Mg and Zn are fueled by H2O2 when it is not the case. It can be confusing.

-        In the conclusions, I would also highlight the limitations of the impact of salts on the motility of most chemically propelled motors that do not utilize bubbling propulsion, an aspect that needs also solutions or new strategies.

Author Response

Reviewer 2 Report (New Reviewer)

Comments and Suggestions for Authors

This manuscript provides an overview of recent advancements in micro/nanomotors (MNMs) for environmental applications. The work has the potential to attract a wide audience in the field of materials, environmental science and robotics. However, to enhance the quality and accuracy of the manuscript, the authors should consider the following points:

1.        The categorization of MNMs based on self-thermophoresis as "Fuel Propulsion" is inaccurate. Please reconsider this classification.

2.        The manuscript should include a discussion of biohybrid MNMs, which are powered by biological forces. This important category of MNMs is currently missing from the review.

3.        Figures 6-9 appear distorted due to changes in their original width or height. Please revise these figures to ensure optimal presentation.

4.        The term "phototropism" is not appropriate to describe the motion of MNMs towards or away from light sources. The correct term is "phototaxis." The first example of the MNMs with phototaxis was reported by Prof. Guan's group (Adv. Mater. 2017, 29, 1603374.).

5.        Please review the manuscript for inappropriate expressions, such as the subtitle "external-field-driven."

6.        Ensure consistency in reference formatting, such as author names and citation styles.

7.        I found some related references (https://doi.org/10.34133/research.0044; ACS Nano 2024, 18, 29248; Sensors and Actuators B: Chemical 2024, 412, 135794; Research 2022, 2022, 9816562; Natl. Sci. Rev. 2021, 8, nwab066; ACS Appl. Mater. Interfaces 2019, 11, 16639).

Author Response

This manuscript is a resubmission of an earlier submission. The following is a list of the peer review reports and author responses from that submission.

Round 1

Reviewer 1 Report

Comments and Suggestions for Authors

Abstract: I consider that the abstract is describing basic things. The abstract should introduce the important of this review in comparison with the already published ones. What do the authors offer?

Introduction: The graphical abstract is similar to the previous reviews previously published such as (among others): 

- DOI: 10.1007/s40820-022-00988-1

-DOI: 10.1021/jacs.8b05762

- DOI: 10.1016/j.nantod.2024.102212

Manuscript: 

- What does Box 1 mean in the middle of the text? it does not have sense.

- Figures: Quality of figure is very poor in general, white gaps, the type of font is different or repeated, the quality of the images is bad. When you take an image of an article, do not reduce it on one of its axes and not the other, because they look bad.

-References: The references must be updated, several works have been forgotten, such as

10.1021/acsnano.4c02115

10.1002/smll.202107619

10.1021/acsami.4c06672

Comments on the Quality of English Language

The english is comprehensible. However, the same grammatical sentences are repeated continously and it is easy to find grammatical mistakes and typos. 

Author Response

After receiving your revisions, I have made the following changes to the article after much consideration. It is worth noting that due to the length of the article and the number of changes, I have used a different background color to highlight the changes made for ease of checking.

Referee: 1

  1. Abstract: l consider that the abstract is describing basic things. The abstract should introducethe important of this review in comparison with the already published ones. What do the authorsoffer?

Abstract as the basic thing that describes the article, in the previous writing process, we didn't describe the purpose of this article and the difference between it and other articles, so after integrating the content of the article, we rewrote the content of the abstract. The specific changes are as follows:

The advent of self-propelled micro/nanomotors represents a paradigm shift in the field of environmental remediation, offering a significant enhancement in the efficiency of conventional operations through the exploitation of the material phenomenon of active motion. Despite the considerable promise of micro/nanomotors for applications in environmental remediation, there has been a paucity of reviews that have focused on this area. This review identifies the current opportunities and challenges in utilising micro/nanomotors to enhance contaminant degradation and removal, accelerate bacterial death or enable dynamic environmental monitoring. It illustrates how mobile reactors or receptors can dramatically increase the speed and efficiency of environmental remediation processes. These studies exemplify the wide range of environmental applications of dynamic micro/nanomotors associated with their continuous motion, force and function. Finally, the review discusses the challenges of transferring these exciting advances from the experimental scale to larger-scale field applications. The revised content has been highlighted with a yellow background.

  1. ntroduction: The graphical abstract is similar to the previous reviews previously published suchas(among others):

The graphical abstract of this paper was designed after our systematic research in the field of environmental remediation of micro/nanomotors, based on the driving mechanisms and pollutant removal mechanisms. Although it is similar in form to some articles, I made sure that it is original and correct.

  1. What does Box 1 mean in the middle of the text? it does not have sense.

The purpose of BOX 1 is that before presenting the environmental applications of MNMs for the detection and removal of pollutants and accelerated bacterial death, it is first necessary to systematically explain the general components of MNMs environmental remediation systems and the extrinsic factors that affect their efficiency. After considering the rationality of the structure of the article, we have changed this part. The revised content has been highlighted with a yellow background.

  1. Figures: Quality of figure is very poor in general, white gaps, the type of font is different orrepeated, the quality of the images is bad. When you take an image of an article, do not reduce iton one of its axes and not the other, because they look bad.

For questions about the pictures in the article. Previously, the quality and clarity of the article images were severely degraded during the layout process. These issues have now been resolved. The revised content has been highlighted with a yellow background.

  1. References: The references must be updated, several works have been forgotten.

In order to make the article more complete, we have added some important literature to the article:

  • Ussia M, Urso M, Oral C M, et al. Magnetic Microrobot Swarms with Polymeric Hands Catching Bacteria and Microplastics in Water[J]. ACS nano, 2024, 18(20): 13171-13183.
  • Vilela D, Guix M, Parmar J, et al. Micromotor‐in‐Sponge Platform for Multicycle Large‐Volume Degradation of Organic Pollutants[J]. Small, 2022, 18(23): 2107619.
  • Asunción Nadal V, Solano Rodríguez E, Jurado Sánchez B, et al. Photophoretic MoS2-Fe2O3 Piranha Micromotors for Collective Dynamic Microplastics Removal[J]. 2024.

The revised content has been highlighted with a yellow background.

Reviewer 2 Report

Comments and Suggestions for Authors

The review discussed application of Micro/Nanomotors in Environmental Remediation. The article can be considered for publication in "Micromachines" after revising the following questions. The comments are below.

1) The author seems to like adding black boxes when drawing pictures. In fact, too many black boxes can create a noticeable sense of fragmentation. We do not recommend adding extra black boxes in some diagrams that involve principles, such as figures 2,3,4.

2) This article is terrible in the citations of figures. Many of thefigures are out of proportion due to stretching. Please quote based on the scale of the original image.

3) Authors should further elaborate on the novelty and importance of the research areas reviewed in this paper compared to existing literature. Although micro/nanomotors are an active area of research, authors need to address the unique contributions of this review.

4) In the third part of the paper, most of the studies discussed by the author are the research progress from 2012 to 2017. We recommend quoting the latest research results to ensure the timeliness of the content of the article and, where possible, comparative analysis with current research trends and findings.

Nano letters, DOI: https://doi.org/10.1021/acs.nanolett.4c03120

Journal of Hazardous Materials, DOI: https://doi.org/10.1016/j.jhazmat.2024.133654

Micromachine, DOI: https://doi.org/10.3390/mi12111380

5) The authors provide application cases of micro/nanomotors in environmental remediation, but the discussion section seems superficial. We recommend exploring in depth the potential impacts, limitations, and possible ways to improve these applications.

Author Response

After receiving your revisions, I have made the following changes to the article after much consideration. It is worth noting that due to the length of the article and the number of changes, I have used a different background color to highlight the changes made for ease of checking.

Referee: 2

  1. The author seems to like adding black boxes when drawing pictures. In fact, too many blackboxes can create a noticeable sense of fragmentation. We do not recommend adding extra blackboxes in some diagrams that involve principles, such as figures 2,3,4.

In order to make the image look more complete, we have eliminated the black border. The revised content has been highlighted with a yellow background.

  1. This article is terrible in the citations of figures. Many of thefigures are out of proportion due tostretching.Please quote based on the scale of the original image.

In order to improve the quality of the original images in the article and make them more enjoyable to look at. We have rearranged the layout of the images in the article. The revised content has been highlighted with a yellow background.

  1. Authors should further elaborate on the novelty and importance of the research areas reviewedin this paper compared to existing literature. Although micro/nanomotors are an active area ofresearch, authors need to address the unique contributions of this review.

In the previous contents of the abstract, we did not highlight the importance of this paper. It has now been revised as described below:

The advent of self-propelled micro/nanomotors represents a paradigm shift in the field of environmental remediation, offering a significant enhancement in the efficiency of conventional operations through the exploitation of the material phenomenon of active motion. Despite the considerable promise of micro/nanomotors for applications in environmental remediation, there has been a paucity of reviews that have focused on this area. This review identifies the current opportunities and challenges in utilising micro/nanomotors to enhance contaminant degradation and removal, accelerate bacterial death or enable dynamic environmental monitoring. It illustrates how mobile reactors or receptors can dramatically increase the speed and efficiency of environmental remediation processes. These studies exemplify the wide range of environmental applications of dynamic micro/nanomotors associated with their continuous motion, force and function. Finally, the review discusses the challenges of transferring these exciting advances from the experimental scale to larger-scale field applications. The revised content has been highlighted with a yellow background.

  1. In the third part of the paper, most of the studies discussed by the author are the researchprogress from 2012 to 2017. We recommend quoting the latest research results to ensure thetimeliness of the content of the article and, where possible, comparative analysis with currentresearch trends and findings.

In order to make the article more complete, we have added some important literature to the article:

  • Zhang T, Ren H, Qin H, et al. Light-Armed Nitric Oxide-Releasing Micromotor In Vivo[J]. Nano Letters, 2024.
  • Zhang X, Chen L, Fu L, et al. Dual-functional metal-organic frameworks-based hydrogel micromotor for uranium detection and removal[J]. Journal of Hazardous Materials, 2024, 467: 133654.
  • Yin B, Wan X, Qian C, et al. Enzyme method-based microfluidic chip for the rapid detection of copper ions[J]. Micromachines, 2021, 12(11): 1380.

The revised content has been highlighted with a yellow background.

  1. The authors provide application cases of micro/nanomotors in environmental remediation, butthe discussion section seems superficial. We recommend exploring in depth the potentialimpacts,limitations,and possible ways to improve these applications

In order to go a step further and provide some suggestions for the environmental remediation field, we have taken a look at the opportunities and challenges facing the micro-nano industry and provided some possible solutions:

For example, (i) The preparation process for some multi-MNMs may involve complex synthesis steps, which can pose challenges for scaling up and commercial production. To overcome these difficulties, advanced fabrication techniques such as laser-based 3D printing can be developed to simplify the preparation steps. (ii) Some MNMs may have poor and irreversible sensitivity, which may reduce the selectivity of MNMs in highly complex environments. To improve the responsiveness of MNMs, further analysis of their response mechanism and optimization of their structural design is necessary. (iii) To better adapt MNMs for practical applications, it is necessary to improve their mechanical properties while maintaining their existing deformable structure. This can be achieved by enhancing the crosslink density and adjusting the hybrid ratios of MNMs. (iv) In terms of load capacity, the load capacity of individual MNMs is very limited due to the small size of individual MNMs. To solve this problem, after exploring the interactions between individual MNMs, the MNMs are enabled to move in a swarm manner by using the three principles of velocity matching, cohesion and consistency. In addition, there is a further need to constrain MNMs to a stable swarm motion pattern so that they can perform complex tasks more efficiently. (v) MNMs are usually made from a variety of materials with complex compositions, so degradability is also a factor to consider. To overcome this obstacle, the use of degradable materials should be the first consideration in their manufacture. In addition, for MNMs made of non-biodegradable materials, appropriate recycling methods should be established to ensure that they do not harm the environment. The revised content has been highlighted with a yellow background.

Round 2

Reviewer 2 Report

Comments and Suggestions for Authors

The authors have solved my problem. I suggest accepting this article.

Author Response

Greetings, esteemed editor:

After receiving your revisions, I have made the following changes to the article after much consideration. It is worth noting that due to the length of the article and the number of changes, I have used a different background color to highlight the changes made for ease of checking.

  1. This review has so much similarity with the previous reviews, can not be distingiushed from other reviews.

In order to highlight the difference between this article and the previous ones, and to give more prominence to the recent advances of MNMs in the environmental field, we have added some difficulties that MNMs may face during environmental remediation and proposed corresponding solutions in the conclusion section. The specific changes are as follows(the changes have been highlighted using a cyan background):

It highlights how mobile reactors or receptors can significantly enhance the effectiveness and speed of environmental remediation and monitoring processes. The innovative features of synthetic MNMs that drive these promising environmental applications are examined. Additionally, the article discusses the current opportunities and challenges associated with utilizing these functional MNMs to improve various environmental solutions. 

(vi) Challenges of safety and biocompatibility: MNMs may pose unknown long-term impacts on ecosystems and human health. To deal with this situation, We should develop biodegradable or bioinspired materials, enabling the robots to safely degrade after completing their tasks, reducing environmental risks. (vii) Challenges of swarm control and coordination: Controlling and coordinating large numbers of MNMs in complex environments, while adapting to unpredictable conditions, is challenging. To overcome this obstacle,We need use artificial intelligence and machine learning algorithms to enhance autonomous decision-making and achieve efficient swarm collaboration. (vii) Challenges of retrieval and environmental accumulation: After completing their tasks, MNMs may be difficult to retrieve, potentially causing environmental pollution. Therefore, we need to employ magnetic recovery systems or design self-decomposing MNMs to ensure they do not leave residues in the environment.

  1. Besides, not only in the paper, but also in the milstone scheme, quite a lot of representative progress are missing, like enzyme powered motors for environmental remediation, contaminants fueled MNMs, , etc., which is the most flaw in this paper.

To make the content of the article more complete, after careful consideration, we have added some new material: examples of environmental remediation using enzymes or pollutants as chemical fuels. The specific changes are as follows (the revised content is highlighted with a cyan background).

Furthermore, in addition to the means by which externally applied fuels and externally applied fields can enable environmental remediation of MNMs, naturally occurring contaminant molecules in the environment can also be used as chemical fuels in the process. For example, Mushtaq et al. demonstrated the catalytic degradation of organic compounds using the magnetoelectric properties of core-shell nanoparticles made from cobalt–bismuth ferrite (CFO-CBO). By integrating the magnetostrictive properties of CFO with the multiferroic nature of BFO, they created MNMs that respond to magnetoelectric stimulation. When subjected to an external magnetic field, these MNMs facilitate the breakdown of organic pollutants through an advanced oxidation process, effectively purifying water. Notably, the process does not require the addition of any sacrificial molecules or external catalysts206.

  1. Mushtaq, F.; Chen, X.; Torlakcik, H.; Steuer, C.; Hoop, M.; Siringil, E.C.; Marti, X.; Limburg, G.; Stipp, P.; Nelson, B.J.; et al. Magnetoelectrically Driven Catalytic Degradation of Organics. Mater. 2019, 31, 1901378.

At the same time, enzymes can be used to advance MNMs for environmental remediation. For example, Orozco et al. introduced a novel micromotor-based water quality testing strategy. This strategy is based on changes in the driving behavior of the artificial biocatalytic microswimmer in the presence of aquatic pollutants156.

  1. Moreover, Figure 9 is not well organized, which should be improved

In order to make the article images more ornamental, we have rearranged the layout of Figure 9.
